# ADVERSARIAL SAMPLE DETECTION THROUGH NEURAL NETWORK FLOW DYNAMICS

## ABSTRACT

We propose a detector of adversarial samples that is based on the view of residual networks as discrete dynamical systems. The detector tells clean inputs from abnormal ones by comparing the discrete vector fields they follow throughout the network's layers. We also show that regularizing this vector field during training makes the network more regular on the data distribution's support, thus making the network's activations on clean samples more distinguishable from those on abnormal samples. Experimentally, we compare our detector favorably to other detectors using seen and unseen attacks, and show that the regularization of the network's dynamics improves the performance of adversarial detectors that use the internal embeddings as inputs, while also improving the network's test accuracy.

## 1 INTRODUCTION

Neural networks have improved performances on many learning tasks, including image classification. They are however vulnerable to adversarial attacks which modify an image in a way that is imperceptible to the human eye but that fools the network into wrongly classifying the modified image (Szegedy et al. (2013)). These adversarial images transfer between networks (Moosavi-Dezfooli et al. (2017)), can be carried out physically (e.g. causing autonomous cars to misclassify road signs (Eykholt et al. (2018))), and can be generated without access to the network (Liu et al. (2017)). Developing networks that are robust to adversarial samples or accompanied by detectors that can detect them is indispensable to deploying them safely in the real word (Amodei et al. (2016)).

In this paper, we focus on detection of adversarial samples. Networks trained with a softmax classifier produce overconfident predictions even for out-of-distribution inputs (Nguyen et al. (2015)). This makes it difficult to detect such inputs via the softmax outputs. A detector is a system capable of predicting whether an input at test time has been adversarially modified or not. Detectors are trained on a dataset made up of clean and adversarial inputs, after the network has been trained. While simply training the detector on the inputs has been tried, using their intermediate embeddings works better (Carlini & Wagner (2017b)). Detection methods vary by which activations to use and how to process them to extract the features that are fed to the classifier that tells clean samples from adversarial ones.

We make two contributions. First, we propose an adversarial detector that is based on the view of neural networks as dynamical systems that move inputs in space, time represented by depth, to separate them before applying a linear classifier (Weinan (2017)). Our detector follows the trajectory of samples in space, through time, to differentiate clean and adversarial images. The statistics that we extract are the positions of the internal embeddings in space approximated by their norms and cosines to a fixed vector. Given their resemblance to the Euler scheme for differential equations, residual networks (He et al. (2016a;b); Weinan (2017)) are particularly amenable to this analysis. Skip connections and residuals are basic building blocks in many architectures (e.g. EfficientNet (Tan & Le (2019)) and MobileNetV2 (Sandler et al. (2018))), and ResNets and their variants (e.g. WideResNet (Zagoruyko & Komodakis (2016)) and ResNeXt (Xie et al. (2017))) remain competitive (Wightman et al. (2021)). Also, Wu et al. (2020) show an increased vulnerability of residual-type architectures to transferable attacks, precisely because of the skip connections. This motivates the need for a detector that is well adapted to residual-type architectures. But the analysis and implementation can extend immediately to any network where most layers have the same input and output dimensions. We test our detector on adversarial samples generated by eight attacks, on three datasets and networks, comparing it to the reference Mahalanobis detector (Lee et al. (2018)) that we largely outperform.

Our second contribution is to use the transport regularization during training proposed in Karkar et al. (2020) to make the activations of adversarial samples more distinguishable from those of clean samples, thus making adversarial detectors perform better, while also improving generalization. We prove that the regularization achieves this by making the network more regular on the support of the data distribution. This does not necessarily make it more robust, but it will make the activations of the clean samples closer to each other and further from those of abnormal out-of-distribution samples, thus making adversarial detection easier. This is illustrated on a toy 2-dimension example in Figure 1 below. We present the related work in Section 2, the background for the regularization in Section 3, the detector in Section 4.1, the theoretical analysis in Section 4.2, and the experiments in Section 5.

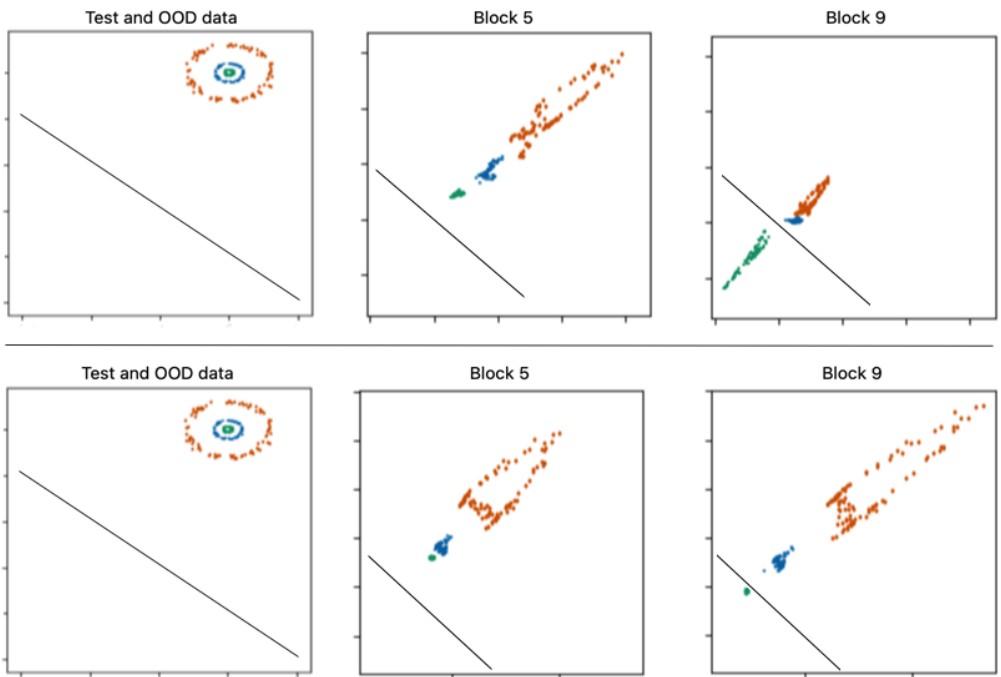

Figure 1: Transformed circles test set from scikit-learn (blue and green) and out-of-distribution points (orange) after blocks 5 and 9 of a small ResNet with 9 blocks. In the second row, we add our proposed regularization during training, which makes the movements of the clean points (blue and green) more similar to each other and more different from the movements of the orange out-of-distribution points than when using the vanilla network in the first row. In particular, without the regularization, the orange points are very close to the clean blue points after block 9 which is undesirable.

## 2    RELATED WORK

Given a classifier $f$ in a classification task and $\epsilon > 0$, an adversarial sample $y$ constructed from a clean sample $x$ is $y = x + \delta$, such that $\|\delta\| \leq \epsilon$ and $f(y) \neq f(x)$. The maximal perturbation size $\epsilon$ has to be so small as to be almost imperceptible to a human. Adversarial attacks are algorithms that find such adversarial samples, and they have been particularly successful against neural networks (Szegedy et al. (2013); Carlini & Wagner (2017a)). We present the adversarial attacks we use in our experiments in Appendix D.1. The main defense mechanisms are robustness, i.e. training a network that is not easily fooled by adversarial samples, and having a detector of these samples.

An early idea for detection was to use a second network (Metzen et al. (2017)). However, this network can also be adversarially attacked. More recent popular statistical approaches include LID (Ma et al. (2018)), which trains the detector on the local intrinsic dimensionality of activations approximated over a batch, and the Mahalanobis detector (Lee et al. (2018)), which trains the detector on the Mahalanobis distances between the activations and a Gaussian fitted to them during training, assuming they are normally distributed. Our detector is not a statistical approach and does not need

batch-level statistics, nor statistics from the training data. Detectors trained in the Fourier domain of activations have also been proposed in Harder et al. (2021). See Aldahdooh et al. (2022) for a review.

Our second contribution is to regularize the network in a way that makes it Hölder-continuous, but only on the data distribution's support. Estimations of the Lipschitz constant of a network have been used as estimates of its robustness to adversarial samples in Weng et al. (2018), Szegedy et al. (2013), Virmaux & Scaman (2018) and Hein & Andriushchenko (2017), and making the network more Lipschitz (e.g. by penalizing an upper bound on its Lipschitz constant) has been used to make it more robust (i.e. less likely to be fooled) in Hein & Andriushchenko (2017) and Cisse et al. (2017). These regularizations often work directly on the weights of the network, therefore making it more regular on all the input space. The difference with our method is that we only endue the network with regularity on the support of the clean data. This won't make it more robust to adversarial samples, but it makes its behavior on them more distinguishable, since they tend to lie outside the data manifold.

That adversarial samples lie outside the data manifold, particularly in its co-dimensions, is a common observation and explanation for why adversarial samples are easy to find in high dimensions (Gilmer et al. (2018); Tanay & Griffin (2016); Song et al. (2018); Ma et al. (2018); Samangouei et al. (2018); Khoury & Hadfield-Menell (2018); Alemany & Pissinou (2022); Feinman et al. (2017)). To the best of our knowledge, Pang et al. (2018) is the only other method that also attempts to improve detection by encouraging the network during training to learn latent representations that are more different between clean and adversarial samples. They do this by replacing the cross-entropy loss by a reverse cross-entropy that encourages uniform softmax outputs among the non-predicted classes. We find that our regularization leads to better classification accuracy and adversarial detection than this method.

The view of ResNets as transport systems was used in Wang et al. (2019) to relate injecting noise into ResNets and ensembling them to approximating the Feynman-Kac convection-diffusion formula, which makes the level sets more regular and the network more robust to adversarial samples.

## 3 BACKGROUND

Our detector is based on the dynamical viewpoint of neural networks that followed from the analogy between residual networks and the Euler scheme made in Weinan (2017). We present this in Section 3.2. The trajectory regularization we use was proposed in Karkar et al. (2020) to improve generalization and we also present it in Section 3.2. The regularity results that this regularization provides for the network require the use of optimal transport theory, which we present in Section 3.1.

### 3.1 OPTIMAL TRANSPORT

Let $\alpha$ and $\beta$ be absolutely continuous densities on a compact set $\Omega \subset \mathbb{R}^d$. The Monge problem is to look for a map $T : \mathbb{R}^d \to \mathbb{R}^d$ moving $\alpha$ to $\beta$, i.e. $T_\sharp \alpha = \beta$, with minimal Euclidean transport cost:

$$\min_{T \text{ s.t. } T_\sharp \alpha = \beta} \int_\Omega \|T(x) - x\|^2 \, \mathrm{d}\alpha(x) \tag{1}$$

and this problem has a unique solution $T^\star$. An equivalent formulation of the Monge problem in this setting is the dynamical formulation. Here, instead of directly pushing points from $\alpha$ to $\beta$ through $T$, we continuously displace mass from time 0 to 1 according to velocity field $v_t : \mathbb{R}^d \to \mathbb{R}^d$. We denote $\phi_t^x$ the position at time $t$ of the particle that was at $x \sim \alpha$ at time 0. This position evolves according to $\partial_t \phi_t^x = v_t(\phi_t^x)$. Rewriting the constraint, Problem (1) is equivalent to the dynamical formulation:

$$\min_v \int_0^1 \|v_t\|^2_{L^2((\phi_t)_\sharp \alpha)} \, \mathrm{d}t \tag{2}$$
$$\text{s.t. } \partial_t \phi_t^x = v_t(\phi_t^x) \text{ for } x \in \text{support}(\alpha) \text{ and } t \in [0,1[, \ \phi_0 = \mathrm{id}, (\phi_1)_\sharp \alpha = \beta$$

### 3.2 LEAST ACTION PRINCIPLE RESIDUAL NETWORKS

A residual network made up of $M$ residual blocks applies $x_{m+1} = x_m + h r_m(x_m)$ for $0 \leq m < M$, with $x_0$ being the input and $h=1$ in practice. The final point $x_M$ is then classified by a linear layer $F$. The dynamic view considers a residual network as an Euler discretization of a differential equation:

$$x_{m+1} = x_m + h r_m(x_m) \ \longleftrightarrow \ \partial_t x_t = v_t(x_t) \tag{3}$$

where $r_m$ approximates the vector field $v_t$ at time $t = m/M$. The dynamic view allows to consider that ResNets are transporting their inputs in space by following a vector field to separate them, the depth representing time, before classification by a linear layer. Karkar et al. (2020) look for a network $F \circ T$ that solves the task while having minimal transport cost:

$$\inf_{T,F} \int_{\Omega} \|T(x) - x\|^2 \, \mathrm{d}\alpha(x) \tag{4}$$
$$\text{subject to} \quad \mathcal{L}(F, T_{\sharp}\alpha) = 0$$

where $T$ is made up of the $M$ residual blocks, $\alpha$ is the data distribution, $F$ is the classification head and $\mathcal{L}(F, T_{\sharp}\alpha)$ is the (cross-entropy) loss obtained from classifying the transformed data distribution $T_{\sharp}\alpha$ through $F$. Given Section 3.1, the corresponding dynamical version of (4) is

$$\inf_{v,F} \int_0^1 \|v_t\|^2_{L^2((\phi_t)_{\sharp}\alpha)} \, \mathrm{d}t \tag{5}$$
$$\text{subject to} \quad \partial_t \phi_t^x = v_t(\phi_t^x) \text{ for } x \in \text{support}(\alpha) \text{ and } t \in [0,1[, \ \ \phi_0 = \text{id}, \ \ \mathcal{L}(F, (\phi_1)_{\sharp}\alpha) = 0$$

Karkar et al. (2020) show that (4) and (5) are equivalent and have a solution such that $T$ is an optimal transport map. In practice, (5) is discretized using a sample $\mathcal{D}$ from $\alpha$ and an Euler scheme, which gives a residual architecture with residual blocks $r_m$ (parametrized along with the classifier by $\theta$) that approximate $v$. This gives the following problem

$$\min_{\theta} \quad \mathcal{C}(\theta) = \sum_{x \in \mathcal{D}} \sum_{m=0}^{M-1} \|r_m(\varphi_m^x)\|^2 \tag{6}$$
$$\text{subject to} \quad \varphi_{m+1}^x = \varphi_m^x + h r_m(\varphi_m^x), \ \ \varphi_0^x = x \text{ for all } x \in \mathcal{D}, \ \ \mathcal{L}(\theta) = 0$$

In practice, we solve Problem (6) using a method of multipliers (see Section 4.2.1). Our contribution is to show, theoretically and experimentally, that this makes adversarial examples easier to detect.

## 4 METHOD

We take the view that a residual network moves its inputs through a discrete vector field to separate them, points in the same class having similar trajectories. Heuristically, for a successful adversarial sample that lies close to clean samples, the vector field it follows has to be different at some step from that of the clean samples, so that it joins the trajectory of the points in another class. In Section 4.1, we present how to detect adversarial samples by looking at these trajectories. In Section 4.2, we discuss the link between the network's regularity and its robustness to adversarial examples and apply the transport regularization by solving Problem (6) to improve detectability of adversarial samples.

### 4.1 DETECTION

Given a residual network that applies $x_{m+1} = x_m + h r_m(x_m)$ to an input $x_0$ for $0 \leq m < M$, we consider the embeddings $x_m$ for $0 < m \leq M$, or the residues $r_m(x_m)$ for $0 \leq m < M$. To describe their positions in space, we take their norms and their cosine similarities with a fixed vector as features to train our adversarial detector on. Using only the norms already gave good detection accuracy. Cosines to other orthogonal vectors can be added to better locate the points at the price of increasing the number of features. We found that using only one vector already gives state-of-the-art detection, so we only use the norms and cosines to a fixed vector of ones. We train the detector (a random forest in practice, see Section 5.2) on these features. The embeddings $x_m$ and the residues $r_m(x_m)$ can equivalently describe the trajectory of $x_0$ in space through the blocks. In practice, we use the residues $r_m(x_m)$, with their norms squared and averaged. So the feature vector given to the random forest for each $x_0$ that goes through a network that applies $x_{m+1} = x_m + h \, r_m(x_m)$ is

$$\left( \frac{1}{d_m} \|r_m(x_m)\|_2^2, \ \cos\left(r_m(x_m), \mathbf{1}_m\right) \right)_{0 \leq m < M} \tag{7}$$

and the label is 0 if $x_0$ is clean and 1 if it is adversarial. Here $\cos$ is the cosine similarity between two vectors and $\mathbf{1}_m$ is a vector of ones of size $d_m$ where $d_m$ is the size of $r_m(x_m)$. For any non-residual

architecture $x_{m+1} = g_m(x_m)$, the vector $x_{m+1} - x_m$ can be used instead of $r_m(x_m)$ on layers that have the same input and output dimension, allowing to apply the method to any network with many such layers. And we do test the detector on a ResNeXt, which does not fully satisfy the dynamic view, as the activation is applied after the skip-connection, i.e. $x_{m+1} = \text{ReLU}(x_m + h\, r_m(x_m))$.

The number of features is twice that of residual blocks (a norm and a cosine per block). This is of the same order as for other popular detectors such as Mahalanobis (Lee et al. (2018)) and LID (Ma et al. (2018)) that extract one feature per residual stage (a residual stage is a group of blocks that keep the same dimension). Even for common large architectures, twice the number of residual blocks is still a small number of features for training a binary classifier (ResNet152 has 50 blocks). More importantly, the features we extract (norms and cosines) are quick to calculate, whereas those of other methods require involved statistical computations on the activations. We include in Appendix D.8 a favorable time comparison of our detector to the Mahalanobis detector. Another advantage is that our detector does not have a hyper-parameter to tune unlike the Mahalanobis and LID detectors.

## 4.2 REGULARIZATION

Regularity of neural networks (typically Lipschitz continuity) has been used as a measure of their robustness to adversarial samples (Weng et al. (2018); Szegedy et al. (2013); Virmaux & Scaman (2018); Hein & Andriushchenko (2017); Cisse et al. (2017)). Indeed, the smaller the Lipschitz constant $L$ of a function $f$ satisfying $\|f(x) - f(y)\| \leq L\|x - y\|$, the less $f$ changes its output $f(y)$ for a perturbation (adversarial or not) $y$ of $x$. Regularizing a network to make it more Lipschitz and more robust has therefore been tried in Hein & Andriushchenko (2017) and Cisse et al. (2017). For this to work, the regularization has to apply to adversarial points, i.e. outside the support of the clean data distribution. Indeed, the Lipschitz continuity obtained though most of these methods and analyses apply on the entire input space $\mathbb{R}^d$ as they penalize the network's weights directly. Likewise, a small step size $h$ as in Zhang et al. (2019b) will have the same effect on all inputs, clean or not.

We propose here an alternative approach where we regularize the network only on the support of the input distribution, making it $\eta$-Hölder on this support (a function $f$ is $\eta$-Hölder on $X$ if $\forall\, a, b \in X$, we have $\|f(a) - f(b)\| \leq C\|a - b\|^\eta$ for some constants $C > 0$ and $0 < \eta \leq 1$, and we denote this $f \in \mathcal{C}^{0,\eta}(X)$). Since this result does not apply outside the input distribution's support, particularly in the adversarial spaces, then this regularity that only applies to clean samples can serve to make adversarial samples more distinguishable from clean ones, and therefore easier to detect. We show experimentally that the behavior of the network will be more distinguishable between clean and adversarial samples in practice in Section 5.1. We discuss the implementation of the regularization in Section 4.2.1 and prove the regularity it endues the network with in Section 4.2.2.

### 4.2.1 IMPLEMENTATION

We regularize the trajectory of the samples by solving Problem (6). This means finding, among the networks that solve the task (condition $\mathcal{L}(\theta) = 0$ in (6)), the network that moves the points the least, that is the one with minimal kinetic energy $\mathcal{C}$. The residual functions $r_m$ we find are then our approximation of the vector field $v$ that solves the continuous version (5) of Problem (6).

We solve Problem (6) via a method of multipliers: since $\mathcal{L} \geq 0$, Problem (6) is equivalent to the min-max problem $\min_\theta \max_{\lambda > 0} \mathcal{C}(\theta) + \lambda\,\mathcal{L}(\theta)$, which we solve, given growth factor $\tau > 0$, and starting from initial weight given to the loss $\lambda_0$ and initial parameters $\theta_0$, through

$$\begin{cases} \theta_{i+1} = \arg\min_\theta \; \mathcal{C}(\theta) + \lambda_i\,\mathcal{L}(\theta) \\ \lambda_{i+1} = \lambda_i + \tau\,\mathcal{L}(\theta_{i+1}) \end{cases} \tag{8}$$

We use SGD for $s > 0$ steps (i.e. batches) for the minimization in the first line of (8), starting from the previous $\theta_i$. When using a ResNeXt, where a residual block applies $x_{m+1} = \text{ReLU}(x_m + r_m(x_m))$, we regularize the norms of the true residues $x_{m+1} - x_m$ instead of $r_m(x_m)$.

### 4.2.2 THEORETICAL ANALYSIS

We first prove a regularity result on each residual $r_m$ that applies on the clean data points moving according to $v$ solution to (5). We take $\Omega \subset \mathbb{R}^d$ convex and compact and the data distribution

$\alpha \in \mathcal{P}(\Omega)$ absolutely continuous and such that $\delta\Omega$ is $\alpha$-negligible. We suppose that there exists an open bounded convex set $X \subset \Omega$ such that $\alpha$ is bounded away from zero and infinity on $X$ and is zero on its complement $X^{\complement}$. From Karkar et al. (2020), we know that problems (4) and (5) are equivalent and have solutions $(T, F)$ and $(v, F)$ such that $T$ is an optimal transport map between $\alpha$ and $\beta := T_{\sharp}\alpha$. We suppose that $\beta$ is absolutely continuous and that there exists an open bounded convex set $Y \subset \Omega$ such that $\beta$ is bounded away from zero and infinity on $Y$ and is zero on $Y^{\complement}$. In the rest of this section, $v$ solves (5) and we suppose that we find a solution to the discretized problem (6) that is an $\varepsilon/2$-approximation of $v$, i.e. $\|r_m - v_{t_m}\|_{\infty} \leq \varepsilon/2$ for all $0 \leq m < M$, with $t_m = m/M$.

**Theorem 4.1.** *For $a, b \in support(\alpha_{t_m})$ where $\alpha_t := (\phi_t)_{\sharp}\alpha$ with $\phi$ solving (5) along with $v$, we have*

$$\|r_m(a) - r_m(b)\| \leq \varepsilon + K\|a - b\|^{\zeta_1} \text{ if } \|a - b\| \leq 1$$
$$\|r_m(a) - r_m(b)\| \leq \varepsilon + K\|a - b\|^{\zeta_2} \text{ if } \|a - b\| > 1$$

*for constants $K > 0$ and $0 < \zeta_1 \leq \zeta_2 \leq 1$,*

*Proof.* The detailed proof is in Appendix C.1. First, $v_t = (T - \mathtt{id}) \circ T_t^{-1}$ where $T_t := (1-t)\mathtt{id} + tT$ and $T$ solves (4). Being an optimal transport map, $T$ is $\eta$-Hölder on $X$. So for all $a, b \in support(\alpha_t)$ and $t \in [0, 1[$, where $\alpha_t = (\phi_t)_{\sharp}\alpha = (T_t)_{\sharp}\alpha$ with $\phi$ solving (5) with $v$, we have

$$\|v_t(a) - v_t(b)\| \leq \|T_t^{-1}(a) - T_t^{-1}(b)\| + C\|T_t^{-1}(a) - T_t^{-1}(b)\|^{\eta} \tag{9}$$

Then, we show that $T_t^{-1}$ is an optimal transport map and so is $\eta_t$-Hölder with $0 < \eta_t \leq 1$. Using also the hypothesis on $r$ and the triangle inequality, we get, for all $a, b \in support(\alpha_{t_m})$

$$\|r_m(a) - r_m(b)\| \leq \varepsilon + C_{t_m}\|a - b\|^{\eta_{t_m}} + CC_{t_m}^{\eta}\|a - b\|^{\eta\eta_{t_m}} \tag{10}$$

Finally, take $K := \max_m C_{t_m} + CC_{t_m}^{\eta}$, $\zeta_1 := \eta \min_m \eta_{t_m}$ and $\zeta_2 := \max_m \eta_{t_m}$. $\square$

We now look at the activations for inputs that are not necessarily in $X$. For inputs $a_0$ and $b_0$, the intermediate embeddings are $a_{m+1} = a_m + hr_m(a_m)$ and $b_{m+1} = b_m + hr_m(b_m)$, and the residues used to compute features for adversarial detection are $r_m(a_m)$ and $r_m(b_m)$. We therefore want to bound $\|r_m(a_m) - r_m(b_m)\|$ as a function of the input variation $\|a_0 - b_0\|$. This is usually done by multiplying the Lipschitz constants of each residual block up to depth $m$, which leads to an overestimation (Huster et al. (2019)), or through more complex estimation algorithms (Virmaux & Scaman (2018); Latorre et al. (2020); Bhowmick et al. (2021)). Bound (9) allows to avoid multiplying the Hölder constants of the block through $T_t^{-1}$. We have the following result (proof in Appendix C.2), which shows the increased regularity guarantee when the inputs $a_0$ and $b_0$ are in $X$:

**Theorem 4.2.** *For all $a_0, b_0 \in X$, we have, for constants $C, L > 0$, that*

$$\|r_m(a_m) - r_m(b_m)\| \leq \varepsilon + \|a_0 - b_0\| + C\|a_0 - b_0\|^{\eta} + L(\|a_m - \phi_{t_m}^{a_0}\| + \|b_m - \phi_{t_m}^{b_0}\|) \tag{11}$$

*The term $\|a_m - \phi_{t_m}^{a_0}\|$ (likewise $\|b_m - \phi_{t_m}^{b_0}\|$) is the distance between the point $a_m$ after $m$ residual blocks and the point $\phi_{t_m}^{a_0}$ we get by following the vector field $v$ up to time $t_m$. Since (9) applies only on the data support, an extra term has to be introduced if $a_0$ is not in $X$ (see the proof). Under more regularity hypotheses on $v$, it is possible to bound these terms. Indeed, if $v$ is $\mathcal{C}^1$ and Lipschitz in $x$ (this is not stronger than the regularity we get on $v$ through our regularization, because it does not give something similar to bound (9)), we have, for constants $R, S > 0$, and for all $a_0, b_0 \in \mathbb{R}^d$, that*

$$\|r_m(a_m) - r_m(b_m)\| \leq \varepsilon + LS\varepsilon + LSRh + \|a_0 - b_0\| + C\|a_0 - b_0\|^{\eta} + LS(dist(a_0, X) + dist(b_0, X))$$

## 5 EXPERIMENTS

We evaluate our method on adversarial samples found by 8 attacks. The threat model is as follows. We use 6 white-box attacks that can access the network and its weights and architecture but not its training data: FGM (Goodfellow et al. (2015)), BIM (Kurakin et al. (2017)), DF (Moosavi-Dezfooli et al. (2016)), CW (Carlini & Wagner (2017a)), AutoAttack (AA) (Croce & Hein (2020a)) and the Auto-PGD-CE (APGD) variant of PGD (Madry et al. (2018)), and 2 black-box attacks that only query the network: HSJ (Chen et al. (2020)) and BA (Brendel et al. (2018)). We present these attacks in Appendix D.1. We assume the attacker has no knowledge of the detector and use the untargeted (i.e.

not trying to direct the mistake towards a particular class) versions of the attacks. We use a maximal perturbation of $\epsilon=0.03$ for FGM, APGD, BIM and AA. We use the $L_2$ norm for CW and HSJ and $L_\infty$ for the other attacks. We compare our detector (which we call the Transport detector or TR) to the Mahalanobis detector (MH) of Lee et al. (2018), and our regularization to reverse cross entropy training of Pang et al. (2018), which is also meant to improve detection of adversarial samples. We use ART (Nicolae et al. (2018)) and Croce & Hein (2020a) to generate the adversarial samples.

We use 3 networks and datasets for experiments: ResNeXt50 on CIFAR100, ResNet110 on CIFAR10 and WideResNet on TinyImageNet. Each network is trained normally with cross entropy, with the transport regularization added to cross entropy (called a LAP-network for Least Action Principle), and with reverse cross entropy instead of cross entropy (called an RCE-network). For LAP training, we use (8) with $\tau=1$, $s=1$ and $\lambda_0=1$ for all networks. These hyper-parameters are chosen to improve validation accuracy during training not adversarial detection. Training details are in Appendix D.2.

## 5.1 PRELIMINARY EXPERIMENTS

Results show that LAP training improves test accuracy. Vanilla ResNeXt50 has an accuracy of $74.38\%$ on CIFAR100, while the LAP-ResNeXt50 has an accuracy of $77.2\%$. Vanilla ResNet110 has an accuracy of $92.52\%$ on CIFAR10, while the LAP-ResNet110 has an accuracy of $93.52\%$ and the RCE-ResNet110 of $93.1\%$. Vanilla WideResNet has an accuracy of $65.14\%$ on TinyImageNet, while the LAP-WideResNet has an accuracy of $65.34\%$. LAP training is also more stable by allowing to train deep networks without batch-normalization in Figure 4 in Appendix D.4. This increased stability is also shown by the often tighter confidence intervals with LAP training in the experiments.

We see in Figure 2 that LAP training makes the transport cost $\mathcal{C}$ more different between clean and adversarial points. Using its empirical quantiles on clean points allows then to detect samples from some attacks with high recall and a fixed false positive rate, without seeing adversarial samples.

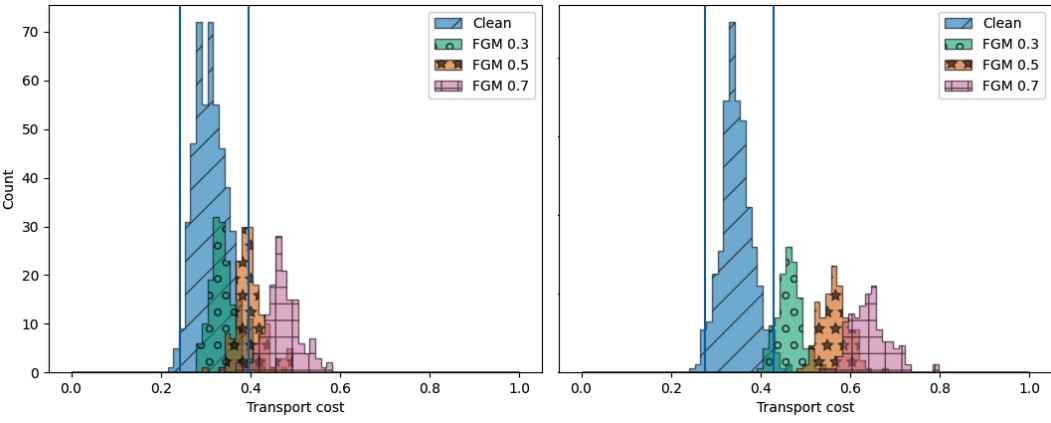

Figure 2: Histogram of transport cost $\mathcal{C}$ for clean and FGM-attacked test samples with different values of $\epsilon$ on CIFAR100. The vertical lines represent the 0.02 and 0.98 empirical quantiles of the transport cost of the clean samples. Left: ResNeXt50. Right: LAP-ResNeXt50.

## 5.2 DETECTION OF SEEN ATTACKS

For detection training, the test set (validation set for TinyImageNet) is split in $0.9/0.1$ proportions into two datasets, B1 and B2. For each image in B1 (respectively B2), an adversarial sample is generated and a balanced detection training set (respectively a detection test set) is created. Since adversarial samples are created for a specific network, this is done for the vanilla version of the network and its LAP and RCE versions. We tried augmenting the detection training dataset with a randomly perturbed version of each image, to be considered clean during detection training, as in Lee et al. (2018), but we found that this does not improve detection accuracy. This dataset creation protocol is standard and is depicted in Figure 3 in Appendix D.3. We tried limiting the datasets to successfully attacked images only as in Lee et al. (2018), but we did not find a significant improvement in detection.

Samples in the detection training set are fed through the network and the features of each detection method are extracted. The classifiers (logistic regression, random forest and SVM from scikit-learn (Pedregosa et al. (2011))) are trained on these features for each detection method and tested on the features extracted from the points in the detection test set. A random forest always works best.

We tried two methods to improve the accuracy of both detectors: class-conditioning and ensembling. In class-conditioning, the features are grouped by the class predicted by the network, and a detector is trained for every class. At test time, the detector trained on the features of the predicted class is used. A detector is also trained on all samples regardless of the predicted class and is used in case a certain class is never targeted by the attack (which might focus on certain classes). We also tried ensembling the class-conditional detector with the general all-class detector: an input is considered an attack if at least one detector says so. In the results below, we report the results of the best performing version, which is often this ensemble of the class-conditional detector and the general detector.

We report the detection accuracy of each detection method on the detection test set for both the vanilla version of the network and its LAP version in Table 1 below. In each cell, the first number corresponds to the vanilla network and the second to the regularized LAP-network. These results are averaged over 5 runs and the standard deviations (which are tight) are in Tables 3, 4, 5 and 6 in Appendix D.5, along with results on RCE-networks and including the FGM and BIM attacks. Since some attacks are slow, we don't test them on all network-dataset pairs in this experiment.

Results in Table 1 show two things. First, our detection method performs better than the Mahalanobis detector, with or without the regularization. Second, both detection methods work better on the LAP versions of the networks most times. The Mahalanobis detector benefits more from the regularization, but on all attacks, the best detector is always the Transport detector. In Tables 3 and 5 in Appendix D.5, RCE often improves detection accuracy in this experiment, but clearly less than LAP training.

Table 1: Average detection accuracy of adversarial samples on Network/LAP-Network.

| Network/Data | Det | Attack | | | | | |
|---|---|---|---|---|---|---|---|
| | | APGD | AA | DF | CW | HSJ | BA |
| ResNet110 | TR | 94.1/**97.5** | 88.9/**94.1** | **99.9**/99.8 | **98.0/98.0** | **97.4/97.4** | 96.6/**97.0** |
| CIFAR10 | MH | 82.1/90.7 | 80.5/90.0 | 91.5/96.7 | 85.6/93.4 | 84.1/92.3 | 80.2/89.6 |
| ResNeXt50 | TR | 96.0/**97.8** | 84.9/**87.6** | **99.8**/99.6 | 95.7/**95.8** | | |
| CIFAR100 | MH | 93.9/94.6 | 83.9/86.6 | 97.3/97.1 | 94.7/94.7 | | |
| WideResNet | TR | 96.0/**96.2** | **81.5**/81.0 | | | | |
| TinyImageNet | MH | 84.2/85.1 | 78.4/78.4 | | | | |

Including the FGM and BIM attacks in Appendix D.5, on CIFAR10, our detector greatly outperforms the MH detector by 9 to 16 percentage points on the vanilla ResNet110. LAP training improves the accuracy of our detector by an average 1.5 points and that of the MH detector by a substantial 8.2 points on average. On CIFAR100, our detector outperforms the MH detector by 1 to 5 points on the vanilla ResNeXt50. LAP training improves the accuracy of both detectors by an average 1 point. On TinyImageNet, our detector greatly outperforms the MH detector by 3 to 12 points on the vanilla WideResNet. LAP training does not change the accuracy of our detector and improves that of the MH detector by 0.6 points on average. Detection rates of successful adversarial samples (i.e. those that fool the network) are much higher than the accuracy on a balanced dataset in Table 1. Even for the AA attack, they are very close to 100% (see Table 10 in Appendix D.7).

## 5.3 DETECTION OF UNSEEN ATTACKS

An important setting is when we don't know which attack might be used or we only have time to train detectors on samples from one attack. We still want our detector to generalize well to samples from unseen attacks. To test this, we use the same vanilla networks as above. The detectors are now trained on the detection training set created by the simplest and quickest attack (FGM) and tested on the detection test sets created by the other attacks. Results are in Table 2 below. We see that our detector has very good generalization to unseen attacks, even those very different from FGM, comfortably

better than the Mahalanobis detector, by up to 19 percentage points in some cases. These results are averaged over 5 runs and the standard deviations are in Tables 7, 8 and 9 in Appendix D.6.

Table 2: Average detection accuracy of samples from unseen attacks after training on FGM.

| Network/Data | Detector | Attack | | | | | | |
|---|---|---|---|---|---|---|---|---|
| | | APGD | BIM | AA | DF | CW | HSJ | BA |
| ResNet110 | TR | **89.32** | **96.02** | **85.10** | **91.02** | **93.18** | **93.00** | **90.92** |
| CIFAR10 | MH | 77.34 | 77.24 | 72.12 | 80.12 | 79.92 | 79.70 | 79.32 |
| ResNeXt50 | TR | **90.82** | **96.12** | **73.28** | **87.60** | **87.54** | **89.90** | **81.70** |
| CIFAR100 | MH | 86.90 | 90.92 | 72.92 | 83.08 | 84.02 | 84.10 | 78.70 |
| WideResNet | TR | **93.26** | **94.66** | **77.04** | **90.62** | **91.42** | | |
| TinyImageNet | MH | 76.96 | 77.02 | 60.36 | 73.18 | 75.52 | | |

However, this experiment shows that our approach has some limitations. We see in Tables 7, 8 and 9 in Appendix D.6 that LAP training does not improve detection accuracy as much anymore, and reduces it in some cases. It improves it for the MH detector on all attacks on ResNet110-CIFAR10 and WideResNet-TinyImageNet by up to 10 points, but not on most attacks on ResNeXt50-CIFAR100. It often reduces it on our detector. But detection of AA samples, the most difficult to detect in Table 2, in particular improves by up to 10 points. LAP training still always does better than RCE training. We claim this limitation is because these training methods reduce the variance of features extracted on the seen attack, thus harming generalization to unseen attacks. This explains why detection of APGD and BIM, variants of FGM, still improves. Fixing this could be an area of future research.

## 5.4 Detection of out-of-distribution samples

Since the analysis applies to all out-of-distribution (OOD) samples, we explore detecting them in a similar setting to the Mahalanobis paper (Lee et al. (2018)): we train a model on a first dataset (CIFAR10), then train detectors to tell this first dataset from a second dataset (CIFAR100 or CW-attacked CIFAR10), then test their ability to tell the first dataset from a third unseen dataset (SVHN). Our detector does very well and better than the MH detector in both experiments we run, and detection accuracy of samples from the unseen distribution is higher than 90%. Details are in Appendix D.9.

## 5.5 Attacking the detector

We consider the case where the detector is also attacked (adaptive attacks). We try 2 attacks on the TR and MH detectors. Both are white-box with respect to the network. The first is black-box with respect to the detector and only knows if a sample has been detected or not. The second has some knowledge about the detector. It knows what features it uses and can attack it directly to find adversarial features. We test these attacks by looking at the percentage of detected successful adversarial samples that they turn into undetected successful adversarial samples. For the first attack, this is 6.8% for our detector and 12.9% for the MH detector on the LAP-ResNet110, and is lowered by LAP training compared to vanilla training. For the second attack it is 14%. Given that detection rates of successful adversarial samples are almost 100% (see Table 10 in Appendix D.7), this shows that an adaptive attack does not circumvent the detector, as detection rates drop to 85% at worst. Details are in Appendix D.10.

## 6 Conclusion

We proposed a method for detecting adversarial samples, based on the dynamical viewpoint of neural networks. The method examines the discrete vector field moving the inputs to distinguish clean and abnormal samples. The detector requires minimal computation to extract the features it needs for detection and achieves state-of-the-art detection accuracy on seen and unseen attacks. We also used a transport regularization that was shown to improve test accuracy to improve the detection accuracy of adversarial detectors. The regularization makes the embeddings closer to each other on the data distribution's support, thus making them more distinguishable from those of abnormal samples.

## REPRODUCIBILITY STATEMENT

Implementation details are in the first paragraph of Section 5 and in Appendix D.1 and D.2. The code is available at `github.com/advadvadvadvadv/adv`, along with weights for the trained ResNet110 models on CIFAR10 that were used.

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

## A    BACKGROUND ON OPTIMAL TRANSPORT

The Wasserstein space $\mathbb{W}_2(\Omega)$ with $\Omega$ a convex and compact subset of $\mathbb{R}^d$ is the space $\mathcal{P}(\Omega)$ of probability measures over $\Omega$, equipped with the distance $W_2$ given by the solution to the optimal transport problem

$$W_2^2(\alpha,\beta) = \min_{\gamma\in\Pi(\alpha,\beta)} \int_{\Omega\times\Omega} \|x-y\|^2 \, \mathrm{d}\gamma(x,y) \tag{12}$$

where $\Pi(\alpha,\beta)$ is the set of probability distribution over $\Omega \times \Omega$ with first marginal $\alpha$ and second marginal $\beta$, i.e. $\Pi(\alpha,\beta) = \{\gamma \in \mathcal{P}(\Omega \times \Omega) \mid \pi_{1\sharp}\gamma = \alpha, \ \pi_{2\sharp}\gamma = \beta\}$ where $\pi_1(x,y) = x$ and $\pi_2(x,y) = y$. The optimal transport problem can be seen as looking for a transportation plan minimizing the cost of displacing some distribution of mass from one configuration to another. This problem indeed has a solution in our setting (see for example Santambrogio (2015); Villani (2008)). If $\alpha$ is absolutely continuous and $\partial\Omega$ is $\alpha$-negligible then the problem in (12) (called the Kantorovich problem) has a unique solution and is equivalent to the following problem, called the Monge problem,

$$W_2^2(\alpha,\beta) = \min_{T \text{ s.t. } T_\sharp\alpha=\beta} \int_\Omega \|T(x)-x\|^2 \, \mathrm{d}\alpha(x) \tag{13}$$

and this problem has a unique solution $T^\star$ linked to the solution $\gamma^\star$ of (12) through $\gamma^\star = (\mathrm{id},T^\star)_\sharp\alpha$. Another equivalent formulation of the optimal transport problem in this setting is the dynamical formulation (Benamou & Brenier (2000)). Here, instead of directly pushing samples of $\alpha$ to $\beta$ using $T$, we can equivalently displace mass, according to a continuous flow with velocity $v_t : \mathbb{R}^d \to \mathbb{R}^d$. This implies that the density $\alpha_t$ at time $t$ satisfies the *continuity equation* $\partial_t\alpha_t + \nabla \cdot (\alpha_t v_t) = 0$, assuming that initial and final conditions are given by $\alpha_0 = \alpha$ and $\alpha_1 = \beta$ respectively. In this case, the optimal displacement is the one that minimizes the total action caused by $v$ :

$$W_2^2(\alpha,\beta) = \min_v \int_0^1 \|v_t\|^2_{L^2(\alpha_t)} \, \mathrm{d}t \tag{14}$$
$$\text{s.t. } \partial_t\alpha_t + \nabla \cdot (\alpha_t v_t) = 0, \ \alpha_0 = \alpha, \alpha_1 = \beta$$

Instead of describing the density's evolution through the continuity equation, we can describe the paths $\phi_t^x$ taken by particles at position $x$ from $\alpha$ when displaced along the flow $v$. Here $\phi_t^x$ is the position at time $t$ of the particle that was at $x \sim \alpha$ at time 0. The continuity equation is then equivalent to $\partial_t\phi_t^x = v_t(\phi_t^x)$. See chapters 4 and 5 of Santambrogio (2015) for details. Rewriting the conditions as necessary, Problem (14) becomes

$$W_2^2(\alpha,\beta) = \min_v \int_0^1 \|v_t\|^2_{L^2((\phi_t)_\sharp\alpha)} \, \mathrm{d}t \tag{15}$$
$$\text{s.t. } \partial_t\phi_t^x = v_t(\phi_t^x), \ \phi_0 = \mathrm{id}, (\phi_1)_\sharp\alpha = \beta$$

and the optimal transport map $T^\star$ that solves (13) is in fact $T^\star(x) = \phi_1^x$ for $\phi$ that solves the continuity equation together with the optimal $v^\star$ from (15). The optimal vector field is related to the optimal map through $v_t^\star = (T^\star - \mathrm{id}) \circ (T_t^\star)^{-1}$, where $T_t^\star = (1-t)\mathrm{id} + tT^\star$ and is invertible. This simply means that the points move in straight lines and with constant speed from $x$ to $T^\star(x)$. The $\mathbb{W}_2(\Omega)$ space is a metric geodesic space, and the geodesic between $\alpha$ and $\beta$ is the curve $\alpha_t$ found while solving (14). It is also given by $\alpha_t = (\pi_t)_\sharp\gamma^\star = (T_t^\star)_\sharp\alpha$, where $\pi_t(x,y) = (1-t)x + ty$. We refer to Section 5.4 of Santambrogio (2015) for these results on optimal transport.

Optimal transport maps have some regularity properties under some boundedness assumptions. We mention the following result from Figalli (2017):

**Theorem A.1.** *Suppose there are $X, Y$, bounded open sets, such that the densities of $\alpha$ and $\beta$ are null in their respective complements and bounded away from zero and infinity over them respectively. Then, if $Y$ is convex, there exists $\eta > 0$ such that the optimal transport map $T$ between $\alpha$ and $\beta$ is $C^{0,\eta}$ over $X$.*
*If $Y$ isn't convex, there exists two relatively closed sets $A, B$ in $X, Y$ respectively such that $T \in C^{0,\eta}(X \setminus A, Y \setminus B)$, where $A$ and $B$ are of null Lebesgue measure.*
*Moreover, if the densities are in $C^{k,\eta}$, then $C^{0,\eta}$ can be replaced by $C^{k+1,\eta}$ in the conclusions above. In particular, if the densities are smooth, then the transport map is a diffeomorphism.*

A final result that we mention is the following, which says that the inverse of the optimal transport map between $\alpha$ and $\beta$ is the optimal transport map from $\beta$ to $\alpha$,

**Theorem A.2.** *If $\alpha$ and $\beta$ are absolutely continuous measures supported respectively on compact subsets $X$ and $Y$ of $\mathbb{R}^d$ with negligible boundaries, then there exists a unique couple $(T, S)$ of functions such that the five following points hold*

- *$T : X \to Y$ and $S : Y \to X$*

- *$T_\#\alpha = \beta$ and $S_\#\beta = \alpha$*

- *$T$ is optimal for the Monge problem from $\alpha$ to $\beta$*

- *$S$ is optimal for the Monge problem from $\beta$ to $\alpha$*

- *$T \circ S \overset{\beta - a.s.}{=} id$ and $S \circ T \overset{\alpha - a.s.}{=} id$*

## B  BACKGROUND ON NUMERICAL METHODS FOR ODEs

We refer to Quarteroni et al. (2007) for this quick background on numerical methods for ODEs. Consider the Cauchy problem $x' = f(t, x)$ with initial condition $x(t_0) = x_0$ and a subdivision $t_0 < t_1 < .. < t_N = t_0 + T$ of $[t_0, t_0 + T]$. Denote the time-steps $h_n := t_{n+1} - t_n$ for $0 \le n < N$ and define $h_{\max} := \max h_n$. In a one-step method, an approximation of $x(t_n)$ is $x_n$ given by

$$\frac{x_{n+1} - x_n}{h_n} = \phi(t_n, x_n, h_n)$$

For $\phi(t, x, h) = f(t, x)$, we get Euler's method: $x_{n+1} = x_n + h_n f(t_n, x_n)$.

**Definition B.1.** *(Consistency and order)* For a one-step method, the *consistency errors* $e_n$, for $0 \le n < N$, are

$$e_n := \frac{x(t_{n+1}) - \Phi(t_n, x(t_n), h_n)}{h_n} = \frac{x(t_{n+1}) - x(t_n)}{h_n} - \phi(t_n, x(t_n), h_n)$$

where $x$ is solution. The *local (truncation) errors* are $h_n e_n$. The method is *consistent* if $\max |e_n|$ goes to zero as $h_{\max}$ goes to zero. For $p \in \mathbb{N}^*$, the method has *order* $p$ if $\max |e_n| \le C h_{\max}^p$ for a constant $C$ that depends on $f, t_0$ and $T$.

**Theorem B.2.** (Consistency criterion) *If $f$ and $\phi$ are continuous then the one-step method is consistent if and only if $\phi(t, x, 0) = f(t, x)$ for all $(t, x)$.*

**Theorem B.3.** (Order criterion) *If $f$ is $\mathcal{C}^p$ and $\phi$ is $\mathcal{C}^p$ in $h$ then the one-step method is of order $p$ if and only if $\partial_h^k \phi(t, x, 0) = \frac{1}{k+1} f^{[k]}(t, x)$ for all $(t, x)$ and $0 \le k < p$ where $f^{[0]} = f$ and $f^{[k]} = \partial_t f^{[k-1]} + f \partial_x f^{[k-1]}$.*

**Corollary B.4.** (Consistency and order of Euler's method) *If $f$ is continuous then Euler's method is consistent. If $f$ is $\mathcal{C}^1$ then Euler's method has order 1.*

**Definition B.5.** *(Zero-stability)* A one-step method is *zero-stable* (or *stable*) if $\exists\, S > 0$ such that for all $(x_n)_{0 \le n \le N}$, $(\tilde{x}_n)_{0 \le n \le N}$ and $(\varepsilon_n)_{0 \le n < N}$ satisfying

$$\frac{x_{n+1} - x_n}{h_n} = \phi(t_n, x_n, h_n) \text{ and } \frac{\tilde{x}_{n+1} - \tilde{x}_n}{h_n} = \phi(t_n, \tilde{x}_n, h_n) + \epsilon_n \tag{16}$$

for $0 \le n < N$, we have $\max_n \|\tilde{x}_n - x_n\| \le S(\|\tilde{x}_0 - x_0\| + T \max_n |\epsilon_n|)$, where $\epsilon_n = \varepsilon_n / h_n$. The constant $S$ is the *stability constant* of the method.

**Theorem B.6.** (Zero-stability criterion) *If $\phi$ is uniformly $L$-Lipschitz in its second variable, then the one-step method is stable with constant $e^{LT}$.*

**Corollary B.7.** (Zero-stability of Euler's method) *If $f$ is Lipschitz in its second variable, then Euler's method is stable.*

**Definition B.8.** *(Convergence)* A numerical method *converges* if its *global error* $\max_n \|x(t_n) - x_n\|$ goes to zero as $h_{\max}$ goes to zero.

**Theorem B.9.** (Convergence criterion) *If a method is consistent and stable with stability constant $S$, then it converges and $\max_n \|x(t_n) - x_n\| \le ST \max |e_n|$. If the method is of order $p$ with constant $C$, then $\max_n \|x(t_n) - x_n\| \le STCh_{max}^p$*

**Corollary B.10.** (Convergence of Euler's method) *Euler's method converges if $f$ is $\mathcal{C}^0$ and Lipschitz in $x$. If $f$ is also $\mathcal{C}^1$ then it converges with speed $O(h_{max})$.*

## C PROOFS

### C.1 PROOF OF THEOREM 4.1

*Proof.* A solution $v$ to (5) exists and is linked to an optimal transport map $T$ that is a solution to (4) through $v_t = (T - \texttt{id}) \circ T_t^{-1}$ where $T_t := (1 - t)\texttt{id} + tT$ which is invertible (see Appendix A).

By Theorem A.1 in Appendix A, being an optimal transport map, $T$ is $\eta$-Hölder on $X$. So for all $a, b \in$ support$(\alpha_t)$ and $t \in [0, 1[$, where $\alpha_t = (\phi_t)_\sharp \alpha = (T_t)_\sharp \alpha$ with $\phi$ solving (5) with $v$, we have

$$\|v_t(a) - v_t(b)\| \leq \|T_t^{-1}(a) - T_t^{-1}(b)\| + C\|T_t^{-1}(a) - T_t^{-1}(b)\|^\eta \tag{17}$$

Since $(\alpha_t)_{t=0}^1$ is a geodesic between $\alpha$ and $\beta = \alpha_1 = T_\sharp \alpha$, then $(\alpha_s)_{s=0}^t$ is a geodesic between $\alpha$ and $\alpha_t$ (modulo reparameterization to $[0, 1]$). And since $\alpha_s = (T_s)_\sharp \alpha$, the map $T_t$ is an optimal transport map between $\alpha$ and $\alpha_t$. Therefore its inverse $T_t^{-1}$ is an optimal transport map (see Theorem A.2 in Appendix A) and is $\eta_t$-Hölder with $0 < \eta_t \leq 1$ (being a push-forward by $T_t$, the support of $\alpha_t$ satisfies the conditions of Theorem A.1 in Appendix A). Therefore, for all $a, b \in$ support$(\alpha_t)$

$$\|v_t(a) - v_t(b)\| \leq C_t\|a - b\|^{\eta_t} + CC_t^\eta\|a - b\|^{\eta\eta_t} \tag{18}$$

and for all $a, b \in$ support$(\alpha_{t_m})$

$$\|r_m(a) - r_m(b)\| \leq \varepsilon + C_{t_m}\|a - b\|^{\eta_{t_m}} + CC_{t_m}^\eta\|a - b\|^{\eta\eta_{t_m}} \tag{19}$$

by the hypothesis on $r$ and the triangle inequality. Let $K := \max_m C_{t_m} + CC_{t_m}^\eta$, $\zeta_1 := \eta \min_m \eta_{t_m}$ and $\zeta_2 := \max_m \eta_{t_m}$. Then, we have the desired result immediately from (19). □

*Remark C.1.* If the convexity hypothesis on the support $Y$ of the target distribution $\beta$ is too strong, we still get the same results almost everywhere. More precisely, if the set $Y$ such that $\beta$ is bounded away from zero and infinity on $Y$ and is zero on $Y^\complement$ is open and bounded but not convex, then the solution map $T$ is $\eta$-Hölder almost everywhere on $X$ (see Appendix A).

*Remark C.2.* If the distributions $\alpha$ and $\beta$ in Theorem 4.1 are $\mathcal{C}^{k,\eta}$ (i.e all derivatives up to the $k$-th derivative are $\eta$-Hölder), then the optimal transport map $T$ is $\mathcal{C}^{k+1,\eta}$. This means that the more regular the data, the more regular the network we find.

### C.2 PROOF OF THEOREM 4.2

*Proof.* Since $T_t^{-1}(\phi_t^x) = x$, we have for any $a_0, b_0 \in X$ by the triangle inequality

$$\begin{aligned}
\|r_m(a_m) - r_m(b_m)\| &\leq \|r_m(a_m) - r_m(\phi_{t_m}^{a_0})\| + \|r_m(\phi_{t_m}^{a_0}) - v_{t_m}(\phi_{t_m}^{a_0})\| + \\
&\quad + \|v_{t_m}(\phi_{t_m}^{a_0}) - v_{t_m}(\phi_{t_m}^{b_0})\| + \\
&\quad + \|r_m(\phi_{t_m}^{b_0}) - v_{t_m}(\phi_{t_m}^{b_0})\| + \|r_m(b_m) - r_m(\phi_{t_m}^{b_0})\| \\
&\leq \varepsilon + \|a_0 - b_0\| + C\|a_0 - b_0\|^\eta + L(\|a_m - \phi_{t_m}^{a_0}\| + \|b_m - \phi_{t_m}^{b_0}\|)
\end{aligned}$$

where $L = \max_m L_m$ and $L_m$ is the Lipschitz constant of $r_m$ (which is Lipschitz being a composition of matrix multiplications and activations such as ReLU). This is the first bound in the theorem.

In this bound, the term $\|a_m - \phi_{t_m}^{a_0}\|$ (and likewise $\|b_m - \phi_{t_m}^{b_0}\|$) represents the distance between the point $a_m$ we get after $m$ residual blocks (i.e. after $m$ Euler steps using the approximation $r$ of $v$) and the point $\phi_{t_m}^{a_0}$ we get by following the solution vector field $v$ up to time $t_m$. By the properties of the Euler method (consistency and zero-stability, see Corollaries B.4, B.7 and B.10 in Appendix B), under more regularity conditions on $v$, it is possible to bound this term. Indeed, if $v$ is $\mathcal{C}^1$ and $M$-Lipschitz in $x$ (this is not stronger than the regularity we get on $v$ through our regularization, because we still need to use (17)), we have for constants $R, S > 0$,

$$\|\phi_{t_m}^{a_0} - a_m\| \leq \|\phi_{t_m}^{a_0} - \tilde{a}_m\| + \|\tilde{a}_m - a_m\| \leq S\varepsilon + SRh$$

where $\tilde{a}_m$ comes from the Euler scheme with access to $v$ (i.e. $\tilde{a}_{m+1} := \tilde{a}_m + hv_{t_m}(\tilde{a}_m)$ and $\tilde{a}_0 := a_0$), $R$ is the consistency constant of the Euler method and $S$ is its zero-stability constant. Likewise, we get the same bound for $\|b_m - \phi_{t_m}^{b_0}\|$.

If $a_0, b_0 \notin X$, we need to introduce $\hat{a}_0 := \mathrm{Proj}_X(a_o)$ and $\hat{b}_0 := \mathrm{Proj}_X(b_o)$ to apply (17). We now get

$$\|r_m(a_m) - r_m(b_m)\| \leq \varepsilon + \|a_0 - b_0\| + C\|a_0 - b_0\|^\eta + L(\|a_m - \phi_{t_m}^{\hat{a}_0}\| + \|b_m - \phi_{t_m}^{\hat{b}_0}\|)$$

Bounding the terms $\|a_m - \phi_{t_m}^{\hat{a}_0}\|$ and $\|b_m - \phi_{t_m}^{\hat{b}_0}\|$ now gives

$$\|\phi_{t_m}^{\hat{a}_0} - a_m\| \leq \|a_m - \tilde{a}_m\| + \|\tilde{a}_m - \phi_{t_m}^{\hat{a}_0}\| \leq S(\|a_0 - \hat{a}_0\| + \varepsilon) + SRh$$

where $\tilde{a}_m$ now comes from the Euler scheme with access to $v$ that starts at $\hat{a}_0$ (meaning $\tilde{a}_{m+1} := \tilde{a}_m + hv_{t_m}(\tilde{a}_m)$ and $\tilde{a}_0 := \hat{a}_0$). Likewise, we get the same bound for $\|b_m - \phi_{t_m}^{\hat{b}_0}\|$.

Since $\|a_0 - \hat{a}_0\| = \mathrm{dist}(a_0, X)$ and $\|b_0 - \hat{b}_0\| = \mathrm{dist}(b_0, X)$, we get the second bound in the theorem. Note that if we use the stability of the ODE instead of the Euler method to bound $\|a_m - \phi_{t_m}^{\hat{a}_0}\|$ we get the same result. Indeed, if $\tilde{a}_m$ again comes from the Euler scheme with access to $v$ that starts at $a_0$ (meaning $\tilde{a}_{m+1} := \tilde{a}_m + hv_{t_m}(\tilde{a}_m)$ and $\tilde{a}_0 := a_0$), we can write, for some constant $F > 0$

$$\|\phi_{t_m}^{\hat{a}_0} - a_m\| \leq \|a_m - \tilde{a}_m\| + \|\tilde{a}_m - \phi_{t_m}^{a_0}\| + \|\phi_{t_m}^{a_0} - \phi_{t_m}^{\hat{a}_0}\| \leq S\varepsilon + SRh + F\|a_0 - \hat{a}_0\|$$

since

$$\|\phi_{t_m}^{a_0} - \phi_{t_m}^{\hat{a}_0}\| \leq \|a_0 - \hat{a}_0\| + \int_0^{t_m} \|v_s(\phi_s^{a_0}) - v_s(\phi_s^{\hat{a}_0})\| \, \mathrm{d}s$$

$$\leq \|a_0 - \hat{a}_0\| + M \int_0^{t_m} \|\phi_s^{a_0} - \phi_s^{\hat{a}_0}\| \, \mathrm{d}s$$

$$\leq F\|a_0 - \hat{a}_0\|$$

where we get the last line by Gronwall's lemma. $\qquad\square$

## D ADDITIONAL EXPERIMENTS

### D.1 ADVERSARIAL ATTACKS

White-box attacks have access to the network's weights and architecture. The Fast Gradient Method (FGM) (Goodfellow et al. (2015)) takes a perturbation step in the direction of the gradient that maximizes the loss. Projected Gradient Descent (PGD) (Madry et al. (2018)) and the Basic Iterative Method (BIM) (Kurakin et al. (2017)) are iterative versions of FGM. We use the Auto-PGD-CE (Croce & Hein (2020a)) variant of PGD which has an adaptive step size. Two slower but more powerful attacks are DeepFool (DF) (Moosavi-Dezfooli et al. (2016)), which iteratively perturbs an input in the direction of the closest decision boundary, and Carlini-Wagner (CW) (Carlini & Wagner (2017a)), which solves an optimization problem to find the perturbation. AutoAttack (AA) (Croce & Hein (2020a)) is a combination of three white-box attacks (two variants of Auto-PGD (Croce & Hein (2020a)) and the FAB attack of Croce & Hein (2020b)), and of the black-box Square Attack (SA) (Andriushchenko et al. (2020)). Black-box attacks don't have any knowledge about the network and can only query it. We use two such attacks: Hop-Skip-Jump (HSJ) (Chen et al. (2020)), which estimates the gradient direction at the decision boundary, and the Boundary Attack (BA) (Brendel et al. (2018)), which starts from a large adversarial input and moves towards the boundary decision to minimize the perturbation. We use ART (Nicolae et al. (2018)) and its default hyper-parameter values to generate the adversarial samples, except for AA for which we use the authors' original code.

### D.2 IMPLEMENTATION DETAILS

For ResNeXt50 (Xie et al. (2017)) on CIFAR100 (Krizhevsky (2009)), we train for 300 epochs using SGD with a learning rate of 0.1 (divided by ten at epochs 150, 225 and 250), Kaiming initialization, a batch size of 128 and weight decay of 0.0001. For RCE training, the only changes are that the learning rate is 0.05 and the initialization is orthogonal with a gain of 0.05.

For ResNet110 (He et al. (2016a)) on CIFAR10 (Krizhevsky (2009)), we train for 300 epochs using SGD with a learning rate of 0.1 (divided by ten at epochs 150, 225 and 250), orthogonal initialization

with a gain of 0.05, a batch size of 256, weight decay of 0.0001 and gradient clipping at 5. For RCE training, the only change is that we don't use gradient clipping.

For WideResNet (Zagoruyko & Komodakis (2016)) on TinyImageNet, we train for 300 epochs using SGD with a learning rate of 0.1 (divided by ten at epochs 150, 225 and 250), orthogonal initialization with a gain of 0.1, a batch size of 114 and weight decay of 0.0001.

For the magnitude parameter of the Mahalanobis detector, we try all the values tried in their paper for the magnitude and we report the best results.

### D.3 ADVERSARIAL DETECTION TRAINING DATA

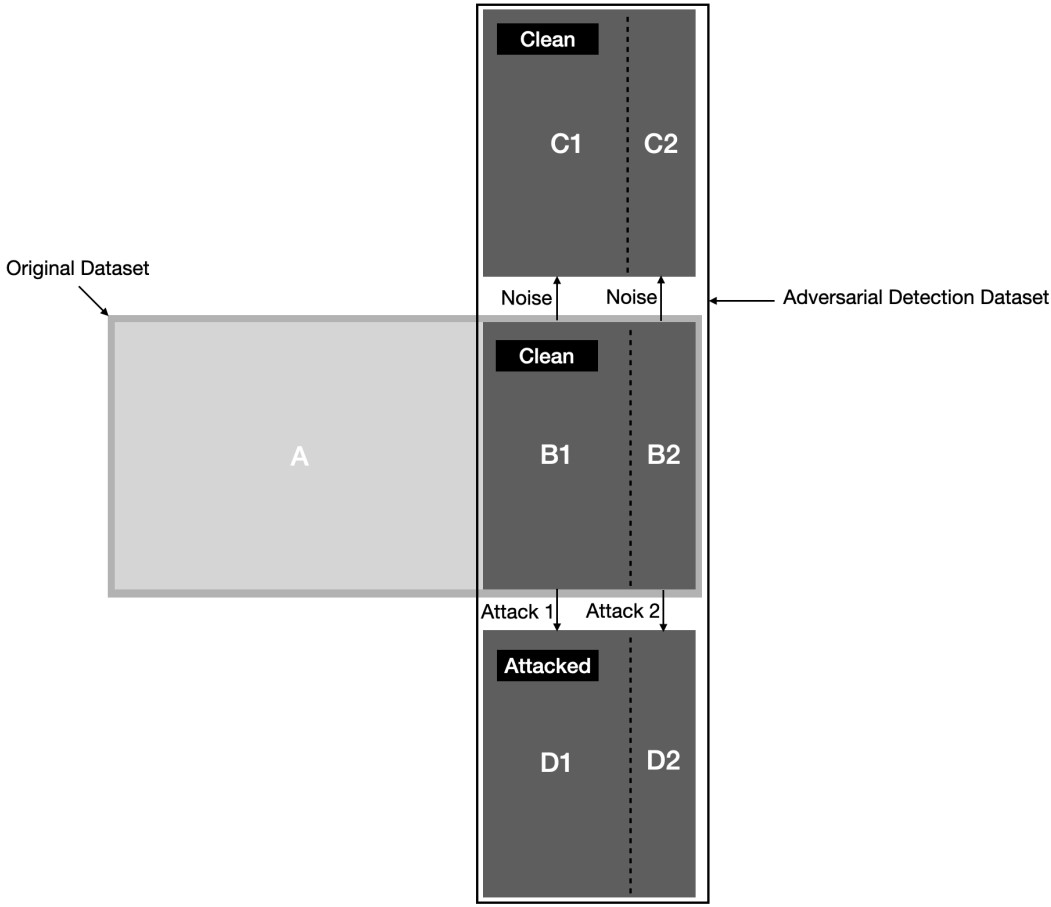

Figure 3: Adversarial detection dataset creation. A∪B1∪B2 is the original dataset, where A is the training set and B1∪B2 is the test set. We create a noisy version of B1∪B2 by adding random noise to each sample in B1∪B2 to get C1∪C2. Noisy samples are considered clean (i.e. not attacked) in adversarial detection training. We create an attacked version of B1∪B2 by creating an attacked image from each image in B1∪B2 to get D1∪D2. In the case of generalization to unseen attacks, Attack 2 used to create D2 from B2 is different from Attack 1 used to create D1 from B1. Otherwise, Attack 1 and Attack 2 are the same. B1∪C1∪D1 is the adversarial detection training set and B2∪C2∪D2 is the adversarial detection test set.

### D.4 PRELIMINARY EXPERIMENTS

We see in Figure 4 below that training deep ResNets without batch-normalization is near impossible, whereas LAP-ResNets maintain the same performance and stability without ResNets for up to 50 blocks. LAP-ResNets are also compared in this regard to the small step method of Zhang et al.

(2019a), which simply adds a small weight $h$ of around $0.1$ in front of the residue function to make ResNets more stable. The Least Action Principle has the same improved stability when training without batch-normalization in Figure 4 as this method, while also improving the test accuracy when batch-normalization is used (Karkar et al. (2020)) which the small step method does not claim.

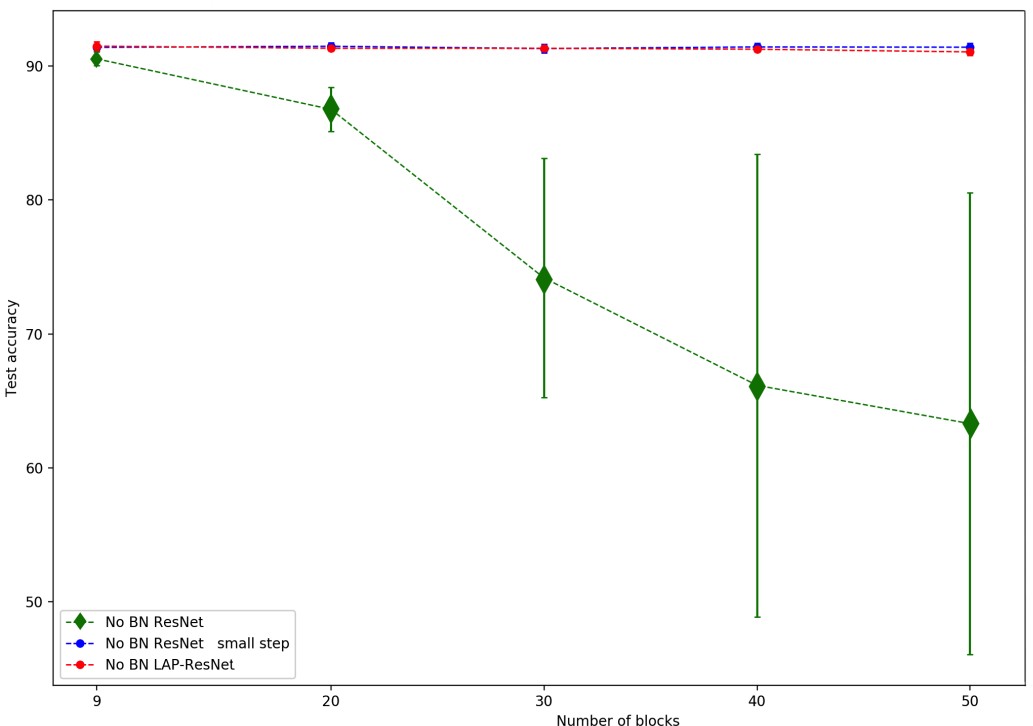

Figure 4: Test accuracy of ResNets of various depths without batch-normalization on CIFAR10.

## D.5 DETECTION OF SEEN ATTACKS

Here, VAN corresponds to detectors trained on a vanilla network, RCE on an RCE-network and LAP on a LAP-network. T corresponds to our detector, and M to the Mahalanobis detector.

Table 3: Average adversarial detection accuracy and standard deviation over 5 runs using ResNet110 on CIFAR10.

| Detector | Attack | | | | |
| | FGM | APGD | BIM | DF | CW |
|---|---|---|---|---|---|
| VAN T | $97.14 \pm 0.59$ | $94.10 \pm 0.41$ | $97.54 \pm 0.52$ | **99.98** $\pm 0.06$ | **98.04** $\pm 0.47$ |
| RCE T | $95.96 \pm 0.54$ | $95.32 \pm 0.80$ | $96.16 \pm 0.61$ | $99.92 \pm 0.10$ | $89.36 \pm 0.53$ |
| LAP T | **98.70** $\pm 0.28$ | **97.50** $\pm 0.37$ | **99.28** $\pm 0.30$ | $99.84 \pm 0.11$ | $97.96 \pm 0.37$ |
| VAN M | $87.78 \pm 4.20$ | $82.08 \pm 4.00$ | $86.78 \pm 4.70$ | $91.50 \pm 3.17$ | $85.58 \pm 2.60$ |
| RCE M | $93.00 \pm 0.60$ | $87.88 \pm 0.49$ | $92.30 \pm 1.03$ | $94.98 \pm 0.40$ | $83.38 \pm 0.49$ |
| LAP M | $95.64 \pm 0.62$ | $90.70 \pm 0.72$ | $95.38 \pm 0.54$ | $96.70 \pm 0.69$ | $93.36 \pm 0.61$ |

Table 4: Average adversarial detection accuracy and standard deviation over 5 runs using ResNet110 on CIFAR10.

| Detector | Attack | | |
|---|---|---|---|
| | AA | HSJ | BA |
| VAN T | $88.88 \pm 1.34$ | $97.38 \pm 0.60$ | $96.56 \pm 0.56$ |
| LAP T | $\mathbf{94.08} \pm 0.80$ | $\mathbf{97.38} \pm 0.43$ | $\mathbf{97.02} \pm 0.18$ |
| VAN M | $80.46 \pm 2.24$ | $84.14 \pm 1.07$ | $80.20 \pm 2.11$ |
| LAP M | $89.96 \pm 0.88$ | $92.30 \pm 0.46$ | $89.62 \pm 0.43$ |

Table 5: Average adversarial detection accuracy and standard deviation over 5 runs using ResNeXt50 on CIFAR100.

| Detector | Attack | | | | | |
|---|---|---|---|---|---|---|
| | FGM | APGD | BIM | AA | DF | CW |
| VAN T | $97.3 \pm 0.5$ | $96.0 \pm 0.5$ | $98.0 \pm 0.3$ | $84.9 \pm 0.7$ | $\mathbf{99.8} \pm 0.2$ | $95.7 \pm 0.4$ |
| RCE T | $97.4 \pm 0.4$ | $96.6 \pm 0.1$ | $97.8 \pm 0.2$ | $50.1 \pm 0.0$ | $99.0 \pm 0.1$ | $92.5 \pm 0.3$ |
| LAP T | $\mathbf{98.3} \pm 0.3$ | $\mathbf{97.8} \pm 0.5$ | $\mathbf{98.9} \pm 0.1$ | $\mathbf{87.6} \pm 0.6$ | $99.6 \pm 0.1$ | $\mathbf{95.8} \pm 0.5$ |
| VAN M | $95.8 \pm 0.5$ | $93.9 \pm 0.5$ | $96.1 \pm 0.6$ | $83.9 \pm 0.7$ | $97.3 \pm 0.4$ | $94.7 \pm 0.7$ |
| RCE M | $96.5 \pm 0.4$ | $94.7 \pm 0.4$ | $96.6 \pm 0.6$ | $50.1 \pm 0.0$ | $97.6 \pm 0.4$ | $88.4 \pm 0.6$ |
| LAP M | $96.8 \pm 0.4$ | $94.6 \pm 0.7$ | $97.8 \pm 0.5$ | $86.6 \pm 0.5$ | $97.1 \pm 0.3$ | $94.7 \pm 0.3$ |

Table 6: Average adversarial detection accuracy and standard deviation over 5 runs using WideResNet on TinyImageNet.

| Detector | Attack | | | |
|---|---|---|---|---|
| | FGM | APGD | BIM | AA |
| VAN T | $\mathbf{95.92} \pm 0.10$ | $96.04 \pm 0.14$ | $96.56 \pm 0.07$ | $\mathbf{81.50} \pm 0.46$ |
| LAP T | $95.86 \pm 0.11$ | $\mathbf{96.16} \pm 0.07$ | $\mathbf{96.56} \pm 0.07$ | $81.04 \pm 0.67$ |
| VAN M | $84.62 \pm 1.32$ | $84.20 \pm 1.32$ | $84.46 \pm 1.29$ | $78.42 \pm 0.88$ |
| LAP M | $85.26 \pm 0.96$ | $85.10 \pm 0.60$ | $85.24 \pm 1.11$ | $78.36 \pm 0.97$ |

## D.6 DETECTION OF UNSEEN ATTACKS

Here, VAN corresponds to detectors trained on a vanilla network, RCE on an RCE-network and LAP on a LAP-network. T corresponds to our detector, and M to the Mahalanobis detector.

Table 7: Average adversarial detection accuracy of unseen attacks after training on FGM and standard deviation over 5 runs using ResNet110 on CIFAR10.

| Detector | Attack | | | | | | |
|---|---|---|---|---|---|---|---|
| | APGD | BIM | AA | DF | CW | HSJ | BA |
| VAN T | $89.3 \pm 1.6$ | $96.0 \pm 0.7$ | $\mathbf{85.1} \pm 1.1$ | $\mathbf{91.0} \pm 0.9$ | $\mathbf{93.2} \pm 1.0$ | $\mathbf{93.0} \pm 0.9$ | $\mathbf{90.9} \pm 0.6$ |
| RCE T | $91.8 \pm 1.1$ | $93.6 \pm 1.1$ | $50.0 \pm 0.1$ | $63.4 \pm 1.1$ | $60.5 \pm 0.9$ | $63.9 \pm 1.0$ | $52.5 \pm 0.5$ |
| LAP T | $\mathbf{92.8} \pm 0.5$ | $\mathbf{98.8} \pm 0.4$ | $84.2 \pm 0.5$ | $75.5 \pm 1.2$ | $75.2 \pm 1.0$ | $76.8 \pm 0.6$ | $75.0 \pm 0.4$ |
| VAN M | $77.3 \pm 4.7$ | $77.2 \pm 4.8$ | $72.1 \pm 3.1$ | $80.1 \pm 3.4$ | $79.9 \pm 3.7$ | $79.7 \pm 3.0$ | $79.3 \pm 3.0$ |
| RCE M | $81.5 \pm 0.6$ | $82.6 \pm 1.3$ | $50.0 \pm 0.1$ | $81.2 \pm 0.7$ | $76.0 \pm 0.9$ | $81.6 \pm 0.9$ | $68.5 \pm 1.2$ |
| LAP M | $87.9 \pm 0.8$ | $84.9 \pm 0.4$ | $81.9 \pm 1.2$ | $81.6 \pm 0.7$ | $81.5 \pm 0.6$ | $81.5 \pm 0.3$ | $81.4 \pm 0.8$ |

Table 8: Average adversarial detection accuracy of unseen attacks after training on FGM and standard deviation over 5 runs using ResNeXt50 on CIFAR100.

| Detector | Attack | | | | | | |
|---|---|---|---|---|---|---|---|
| | APGD | BIM | AA | DF | CW | HSJ | BA |
| VAN T | **90.8** ± 0.5 | 96.1 ± 0.6 | 73.3 ± 0.9 | **87.6** ± 0.2 | **87.5** ± 0.4 | **89.9** ± 2.2 | **81.7** ± 1.6 |
| RCE T | 87.7 ± 0.5 | 95.1 ± 0.9 | 50.0 ± 0.1 | 72.3 ± 0.4 | 61.9 ± 0.5 | 72.4 ± 0.4 | 57.7 ± 0.4 |
| LAP T | 89.3 ± 0.8 | **97.7** ± 0.3 | 74.0 ± 1.3 | 76.0 ± 1.0 | 74.7 ± 1.1 | 78.1 ± 3.4 | 71.9 ± 3.9 |
| VAN M | 86.9 ± 0.1 | 90.9 ± 1.0 | 72.9 ± 0.9 | 83.1 ± 0.7 | 84.0 ± 1.1 | 84.1 ± 3.4 | 78.7 ± 1.4 |
| RCE M | 82.0 ± 0.7 | 88.6 ± 0.8 | 50.0 ± 0.1 | 74.1 ± 0.8 | 63.0 ± 0.6 | 74.6 ± 0.3 | 63.2 ± 0.9 |
| LAP M | 86.7 ± 0.9 | 93.9 ± 0.4 | **80.0** ± 0.6 | 79.4 ± 1.6 | 80.9 ± 2.0 | 80.5 ± 3.7 | 78.2 ± 2.0 |

Table 9: Average adversarial detection accuracy of unseen attacks after training on FGM and standard deviation over 5 runs using WideResNet on TinyImageNet.

| Detector | Attack | | | | |
|---|---|---|---|---|---|
| | APGD | BIM | AA | DF | CW |
| VAN T | 93.26 ± 0.60 | 94.66 ± 0.49 | **77.04** ± 0.74 | **90.62** ± 0.60 | 91.42 ± 1.06 |
| LAP T | **93.48** ± 0.72 | **94.80** ± 0.56 | 76.58 ± 0.48 | 90.12 ± 0.55 | **91.52** ± 0.89 |
| VAN M | 76.96 ± 0.94 | 77.02 ± 1.08 | 60.36 ± 0.62 | 73.18 ± 0.59 | 75.52 ± 0.82 |
| LAP M | 77.96 ± 0.49 | 78.00 ± 0.77 | **61.96** ± 0.89 | 73.98 ± 1.12 | 76.22 ± 0.83 |

## D.7 DETECTION RATE OF SUCCESSFUL ADVERSARIAL SAMPLES

As in Lee et al. (2018), we might be only concerned with detecting adversarial samples that successfully fool the network. We find that the detection rate of successful adversarial samples is always higher than the accuracy on a balanced test set of clean and adversarial samples reported above. On seen attacks, the detection rate of successful adversarial samples is even very close to $100\%$ on all attacks. We report the detection rate of successful adversarial samples in Table 10 below for the AA attack (the most difficult attack to detect in the previous tables), both when it is seen and unseen during detection training. In the unseen case, the training is carried out using adversarial samples generated by the FGM attack.

Table 10: Average detection rates of successful adversarial samples on Network/LAP-Network.

| Network/Data | Det | Attack | |
|---|---|---|---|
| | | AA (seen) | AA (unseen) |
| ResNet110 | TR | **99.9**/**99.9** | **98.5**/80.7 |
| CIFAR10 | MH | 88.3/95.4 | 79.4/81.1 |
| ResNeXt50 | TR | **99.9**/99.7 | 74.0/72.2 |
| CIFAR100 | MH | 98.8/98.7 | 72.8/**86.6** |
| WideResNet | TR | **99.9**/**99.9** | 91.1/**92.5** |
| TinyImageNet | MH | 95.6/96.5 | 71.1/70.1 |

## D.8 TIME COMPARISON

With a ResNeXt50 on CIFAR100 and a Tesla V100 GPU, it takes our method (including the time to generate FGM attacks) 66 seconds to extract its features from both the clean and the adversarial samples, while it takes the Mahalanobis method 110 seconds. Mahalanobis also extracts some

statistics from the training set prior to adversarial detection training, which takes an additional 35 seconds. Our feature vector is of size 32, compared to 5 for the Mahalanobis detector. So our random forest takes only 4 more seconds to train than the Mahalanobis one (7 vs 3 seconds). Computation of the features our detector uses (norms and cosines) is in $O(MD)$, where $M$ is the number of residual blocks and $D$ is the largest embedding dimension inside the network.

### D.9    DETECTION OF OUT-OF-DISTRIBUTION SAMPLES

We test detection of out-of-distribution (OOD) samples in a similar setting to the Mahalanobis paper Lee et al. (2018). We use the same ResNet models trained on CIFAR10. Since the detectors need to be trained, we are in the OOD setting where we have a first dataset for training the network (CIFAR10 here) and a second dataset from another distribution that is not the test OOD distribution to train the detector on. This could be another dataset (CIFAR100 here), some images found in the wild, or a perturbation of our dataset that we generate using an adversarial attack (CW on CIFAR10 here). Detectors can then be used by training them to distinguish between these first two datasets, and then testing them on distinguishing between the first dataset and a third unseen dataset (SVHN (Netzer et al. (2011)) here). Results are in Table 11 below. Our detector performs very well and better than the MH detector in both experiments. Without any extra data available, using an adversarial attack allows to detect OOD samples from an unseen distribution with more than $90\%$ accuracy.

Table 11: Average OOD detection accuracy and standard deviation over 5 runs using ResNet110 trained on CIFAR10.

| | OOD Experiment 1 | | OOD Experiment 2 | |
| Detector | CIFAR100 (seen) | SVHN (unseen) | CW-CIFAR10 (seen) | SVHN (unseen) |
|---|---|---|---|---|
| VAN TR | $98.30 \pm 0.46$ | $97.46 \pm 0.49$ | $\mathbf{97.42} \pm 0.57$ | $\mathbf{91.38} \pm 0.95$ |
| RCE TR | $\mathbf{98.42} \pm 0.40$ | $98.20 \pm 0.39$ | $91.54 \pm 6.06$ | $77.58 \pm 6.72$ |
| LAP TR | $98.30 \pm 0.22$ | $\mathbf{98.50} \pm 0.47$ | $97.28 \pm 0.62$ | $85.46 \pm 2.64$ |
| VAN MH | $86.88 \pm 1.52$ | $91.28 \pm 0.92$ | $81.80 \pm 1.96$ | $83.76 \pm 1.13$ |
| RCE MH | $94.82 \pm 0.45$ | $92.16 \pm 0.57$ | $76.74 \pm 2.75$ | $54.24 \pm 3.46$ |
| LAP MH | $94.84 \pm 0.41$ | $90.46 \pm 1.45$ | $89.68 \pm 0.65$ | $76.72 \pm 1.73$ |

### D.10    ATTACKING THE DETECTOR

We consider here the case where the attacker also attacks the detector (adaptive attacks). We try two such attacks on the TR and MH detectors on ResNet110 trained on CIFAR10. Both attacks are white-box with respect to the network. The first is black-box with respect to the detector. It only knows if an adversarial sample has been detected or not. The second has some knowledge about the detector. It knows what features it uses and can attack it directly to find adversarial features. We test these attacks by looking at the percentage of detected successful adversarial samples that they turn into undetected successful adversarial samples that fool both the network and the detector.

The first attack proceeds as follows. A strong white-box attack (CW or AA) is used on the network on image $x$ that has label $y$. If it finds a successful adversarial image $\tilde{x}$ that fools the network into predicting $\tilde{y} \neq y$ but is detected by the detector, the attacker will attempt to modify this image $\tilde{x}$ so that the network and the detector are both fooled. For this, the image $\tilde{x}$ is used as the initialization for an attack (HSJ) on a black-box Network-Detector system. The attacker considers that the Network-Detector behaves as follows: it outputs the class prediction of the network if the detector does not detect an attack and outputs an additional 'detected' class if the detector detects an attack. The attacker attacks this Network-Detector on image $\tilde{x}$ targeting the $\tilde{y}$ label. This way the network makes a mistake and the 'detected' class is avoided. On the vanilla ResNet110, this attack turns only $8.8\%$ of 1700 detected successful adversarial samples $\tilde{x}$ into undetected successful adversarial samples on our detector, compared to $28.8\%$ on the Mahalanobis detector. These percentages are lower on the LAP-ResNet110 as they drop to $6.8\%$ on our detector and $12.9\%$ on the Mahalanobis detector. This shows that LAP training improves the robustness of both adversarial detectors to being attacked themselves, and that the Transport detector is more robust than the MH detector.

The second attack is very similar to the adaptive attack used in Carlini & Wagner (2017b) to break the Kernel Density detector of Feinman et al. (2017). It proceeds as follows. A strong white-box attack (CW or AA) is used on the network on image $x$ that has label $y$. If it finds a successful adversarial image $\tilde{x}$ that fools the network but is detected by the detector, the detection features $\tilde{z}$ that $\tilde{x}$ generates when run through the network are used as the initialization for a black-box attack (HSJ) on the detector. If successful adversarial detection features $z^*$ that fool the detector are found, the attacker has to find an adversarial perturbation of $x$ that still fools the network and that generates these features $z^*$ (or close features that also fool the detector) when run through the network. We do this as in Carlini & Wagner (2017b) by solving the following optimization problem:

$$\min_{x^*} -L(N(x^*), y) + c_1 \|D(x^*) - z^*\| + c_2 \|x^* - x\| \tag{20}$$

where $L$ is the cross-entropy loss, $N$ is the network, and $D$ is the (differentiable) function that returns the detection features of its input. This optimization problem is differentiable and we try differentiable optimization algorithms such as BFGS and NR to solve it. The initial detected successful adversarial image $\tilde{x}$ is used as initialization as in Carlini & Wagner (2017b). This attack turns $14\%$ of detected successful adversarial samples $\tilde{x}$ into undetected successful adversarial samples on our detector on the LAP-ResNet110.

Given that initial detection rates of successful adversarial samples are almost $100\%$ (see Table 10 in Appendix D.7), this shows that adaptive attacks do not (at least not easily) circumvent the detector, as detection rates drop to $85\%$ at worst. Obviously, the second attack is stronger than the first one, but it can probably still be improved by using for example a white-box attack that is specific to random forests for attacking the detector such as Kantchelian et al. (2016) or Zhang et al. (2020). However, the difficulty of combining the attack on the network with that on the detector remains. It is the non-differentiability of the random forest that forces either this separate treatment of network and detector then the use of a proxy differentiable term for the detector (here $\|D(x^*) - z^*\|$ in (20)) to combine both, or the use of a black-box method as in the first attack. Also, we did not consider here the ensemble of the class-conditional detector and the general detector, which is the best performing version of the detector (see Section 5.2), and should be even more robust to adaptive attacks, as the attacker will have to fool two random forest detectors at once and target a particular label, constraining further the optimization problem he solves.

