# OpenReview forum: "Adversarial Attack Detection Through Network Transport Dynamics"
_ICLR.cc/2023/Conference — Submitted to ICLR 2023_

### Official Review · Reviewer_Xhhn · 2022-10-18

**Confidence:** 5
**Correctness:** 4
**Technical Novelty And Significance:** 3
**Empirical Novelty And Significance:** 3
**Recommendation:** 8

**Clarity, Quality, Novelty And Reproducibility:**

Overall, the presentation of the paper is very good.

The quality of the English text is fair.

Figures and Tables are of good quality.

The topic addressed by the manuscript is source of abundant literature, but still an open issue and relevant for ICLR.

The references are appropriate.

The contribution is significant (at least in the domain of Computer Vision).

The basic idea is intriguing and novel (to the best of my knowledge), and may potentially inspire future works.

The results are reproducible: the datasets are publicly available, and the manuscript provides a link to an (anonymous) repository containing the source-code.

**Strength And Weaknesses:**

STRENGTHS:
+ Strong theoretical analysis
+ Experiments on multiple datasets
+ Repeated trials that allow statistical tests
+ The “runtime” is evaluated, thereby allowing to gauge the overhead of the method
+ Good writing
+ Good presentation
+ Interesting (and novel) idea


WEAKNESSES
- The focus on “security” is not sound
- The evaluation (and the entire method) appears to be exclusively tailored for data in the form of “images”


**Summary Of The Paper:**

The paper tackles the problem of detection of adversarial examples targeting (deep) neural networks. Specifically, the paper models the underlying operations of neural networks as a “dynamic process”, wherein information is propagated “over time” (i.e., goes through layers of neurons). By acting upon such intuition, the paper proposes to detect adversarial examples by analyzing “what happens” in-between each layer, thereby modeling it as an “anomaly detection” problem. The proposed method is grounded in solid theoretical analyses, and is then experimentally evaluated on neural networks trained over diverse benchmark datasets (all of images). The experiments are repeated multiple times, and the overall findings confirm the validity of the proposal, which outperform a reputable prior work (Lee et al. (NeurIPS 2018)). Furthermore, the paper also shows that by “tweaking” the proposed detection method, it is also possible to improve its generalization capabilities---i.e., detecting adversarial examples that do not conform to what the detector is expected to identify. Such a property is also found to be true via comprehensive experiments.

**Summary Of The Review:**

I liked the paper, a lot. I commend the authors for their work: despite not being “groundbreaking”, I consider this paper to be a solid contribution to the state-of-the-art, which deserves to be accepted to ICLR. Even by assuming a very critical eye, I was not able to find any significant flaw in the paper.

Something that I particularly appreciated is reporting the “runtime”: such metric is often forgotten in ML papers, but it is of high importance for practitioners as it allows to gauge the “cost” of a proposed method. Moreover, I also loved Section 5.2: here, the authors describe what I consider to be some “negative results”. I think more papers should include them. Props!

However, I believe that the paper is currently affected by a single issue---which stems from me being a researcher with a “security background,” and hence being critical of this aspect. Although I do not consider such issue to be ground for rejection, I truly invite the authors to fix it. Let me elaborate, and propose actionable means of remediations.

**The “attack” is misleading.**
The paper poses itself as tackling a “security-related” problem, but does not support such security focus in the appropriate way. Indeed, let me quote the first line of the abstract: “Adversarial attacks are perturbations to the input that don’t change its class for a human observer, but fool a neural network into changing its prediction.” The key term, here, is “attack”: from a security standpoint, an “attack” requires an “attacker”, who “does something (malicious) in order to achieve a given goal”. However, the paper does not provide any support for the “attacks” considered in the paper: this is due to the lack of a proper threat model---which is a lack that is becoming endemic in adversarial ML literature (see [A], which reports the opinions of researchers, policy makers and practitioners). To my understanding, the paper (and its contribution) does not deal with the “detection of adversarial attacks”, but rather with the “detection of adversarial examples”. Although the two topics have several similarities, they do not overlap: my take is that the paper addresses the more general problem of “adversarial examples,” which is useful to assess the “robustness” of ML methods and not their overall “security” (I quote the term “false sense of robustness” used in a recent NeurIPS paper [B]).
To address this problem, I invite the authors to revise the paper by replacing “adversarial attacks” with “adversarial examples” whenever necessary. Alternatively, the authors can insert a proper “threat model”: this could be done by moving the current Section 4.2.2 into the Appendix. My stance is that the paper is being oversold as being (also) a "security paper": the authors should prove that I'm wrong, or -- if I'm not -- then they should fix their paper accordingly.

**Dubious utility of “detection”.**
This is a direct consequence of the previous point. The goal of a “detection” mechanism (in the context of  “detecting an attack”) is to act as a “trap” for the attacker. In other words, the detector “expects the attacker to do A, and if they do so, then the attacker falls into the trap and is therefore detected”. However, the lack of a threat model describing what the attacker can/cannot do prevents from determining the utility of the detection mechanism as a security measure. Such a shortcoming further supports my previous observation of shifting the focus of the paper from “detecting adversarial attacks” to “detecting adversarial examples”. I point the authors to [C] for a good paper that truly models the problem of “detection of adversarial *attacks*”.
I acknowledge that the paper reproduces well-known white/black box attacks, but I still do not accept the lack of a threat model. A way to rectify the issue is by mentioning that "we evaluate our method by assuming adversarial examples found by means of the white/black box attacks proposed in (ref)".



**General or Specific?**
Something that is not fully clear to me is whether the proposed idea (which I found very enticing!) can be applied to “any” neural network, or only to residual nets. Let me explain. The paper proposes a method whose main intuition lies in the modeling the “depth” of a neural network as “time”. However, this leads me to think: would this idea “work” for those neural networks that are “time-aware”, such as recurrent neural networks (RNN) and LSTM? This is because, by following the paper’s idea, an input can go “back in time”. The paper does not take such architectures into account, so I am inclined to believe that it does not work. According to the introduction, the paper focuses on “residual networks” because they are “particularly amenable to this analysis”. The authors fairly state that “the analysis and implementation can extend immediately to any network where most layers have the same input and output dimensions.”, but they do not mention whether this also applies to RNN or LSTM; moreover, in Section 3.2, the definitions clearly state “Residual Networks”.
I hence ask the authors to clarify whether the proposed method can cover only “residual networks” (as well as any model that approximates such networks), or not. This should be clearly stated in the Introduction.

**Only Image data.**
This is self-explanatory: although the proposed methodology is (theoretically) applicable to any (residual?) neural network, the paper (and its evaluation) appears to be tailored for applications of neural networks that focus on image analysis. There are many domains in which neural networks are used today, and focusing on a single data-type prevents to assess the general applicability of the proposed technique. A potential fix can be, e.g., performing a proof of concept experiment on a dataset that does not include images---albeit this may be hard to find, since residual nets (which I believe to be “required” for the proposed method) are mostly useful for images.

Some additional issues:

•	I recommend to change the title: as a “security” researcher, I was misled into thinking that the paper dealt with adversarial attacks in the “network traffic analysis” domain (the term “transport” is a key component in networking). Specifically, I endorse the authors to replace the term “transport” with something else. Maybe “updates” is a better fit?

•	The first sentence in Section 5.1 should be better contextualized “First, the regularization improves generalizations.” Instead of "First", it should report "this result shows that..."

•	In Section 4: “use the transport regularization (6) to improve detectability of adversarial attacks.” It is not clear what “(6)” stands for.

•	I was originally taken aback by the “abstract”. In truth, I did not like the abstract at all. I think the authors tried to blend “technical” details with “simple” jargon, but the result (imho) is something that does not give true credit to the paper’s value.

EXTERNAL REFERENCES

[A]: Apruzzese, G., Laskov, P., de Oca, E. M., Mallouli, W., Rapa, L. B., Grammatopoulos, A. V., & Franco, F. D. (2022). The Role of Machine Learning in Cybersecurity. Digital Threats: Research and Practice.

[B]: Pintor, M., Demetrio, L., Sotgiu, A., Manca, G., Demontis, A., Carlini, N., ... & Roli, F. (2022). Indicators of attack failure: Debugging and improving optimization of adversarial examples. NeurIPS 2022.

[C]: Shan, S., Wenger, E., Wang, B., Li, B., Zheng, H., & Zhao, B. Y. (2020, October). Gotta catch'em all: Using honeypots to catch adversarial attacks on neural networks. In Proceedings of the 2020 ACM SIGSAC Conference on Computer and Communications Security (pp. 67-83

---

> ### Author Response · Authors · 2022-11-10
> **Answer to Reviewer Xhhn: Threat model and extensions**
>
> Thank you very much for your review.
>
> Changes in the paper (they are in blue): please note that tables 1 and 2 and sections 5.2.1 and 5.2.2 (white-box and black-box attacks) have been merged to provide more space. Also a comparison with RCE (a training method that aims to improve detectability of adversarial samples) has been added to the tables in the appendix. We verify that this method does often improve detection, but always less than our regularization. Experiments with positive results on AutoAttack and OOD detection have also been added. The code in the anonymous github has been updated to include the new experiments.
>
> A. "The “attack” is misleading", "Dubious utility of “detection”":
>
> Indeed we used the terms adversarial attack and adversarial sample rather interchangeably, and we agree that speaking of detecting adversarial samples is more appropriate. We modified the paper so as to only speak of detecting adversarial samples, an attack being the algorithm that maliciously generates such samples. We included a clear summary of the threat model in the beginning of section 5: 6 attacks are white-box (have access to the network and its weights and architecture, but not to the training data), and 2 are black-box (can only query the network). All don't have access to or knowledge of the detector. Thank you for the reference to the very nice paper [C].
>
> B. "General or Specific?":
>
> The method is not theoretically nor implementation-wise restricted to residual networks. As we mention in 4.2.1, the analysis and implementation still apply to any network as soon as it has some layers that keep the same input and output dimension and so for which a residue can be computed and which can therefore be re-written as residual blocks (through $r_m(x) = g_m(x_m) - x_m$ if the layer is $x_{m+1} = g_m(x_m)$). The presentation is done for resnets for this reason, as other networks will have to be reformulated as residual networks. We included this discussion in section 4.1. The title "Least Action Principle Residual Networks" is only that of section 3.2 which presents the regularization in the setting of residual networks, but we already apply the regularization to ResNeXt which are not strictly speaking residual networks in the sense of section 3.2.
>
> In practice, all the networks we use (other than in the toy example in Figure 1) are already outside the strict theoretical framework, as they include some downsampling layers that change the dimension and are therefore ignored by the detector and the regularization, and the ResNeXt model is not an Euler scheme as the activation is applied after the shortcut (i.e. $x_{m+1} = \text{ReLU}(x_m + h \ r_m(x_m))$). And yet the method works very well on all three networks. Also note that many networks such as EfficientNet and MobileNet are made up in large part of some sort of residual blocks, and so for which the detector and regularization can immediately be applied. This is mentioned in Section 4.1 and in the introduction.
>
> The critical point for extension to other networks is how many layers keep the same dimension and are they or not concentrated in a particular part of the network. As for RNNs and LSTMs, the analogy with time/space is more complicated to make as the time element is in the input data itself, not just in the depth. We do not claim that our method applies as such for recurrent models.
>
> C. "Only Image data":
> Yes, we agree, but we were considering testing the method on detection of out-of-domain samples first, before considering other data types. Residual networks were introduced on the ImageNet challenge and as you say they continued thereafter to be used mostly in computer vision, but the idea of residuals and skip connections is not motivated by computer vision considerations, rather by the goal of avoiding gradient vanishing. It is the convolutions inside the residual blocks that are adapted to image data. Residual networks have therefore been applied to audio data in [1] for example by changing the form of the convolution.
>
>
> As for the additional issues:
>
> We have modified the title and abstract as you suggest. We added the word neural before networks and replaced transport by flow. We hope this alleviates any confusion.
>
> We have also corrected the other two points (the first sentence in section 5.1 and the reference to Problem (6) in section 4) in the updated version of the paper.
>
>
> [1] ERANNs: Efficient Residual Audio Neural Networks for Audio Pattern Recognition, Sergey Verbitskiy, Vladimir Berikov, Viacheslav Vyshegorodtsev, Pattern Recognition Letters, 2022

---

> > ### Comment · Reviewer_Xhhn · 2022-11-12
> > **Ack**
> >
> > I thank the authors for their response and for the abundant changes and improvements!
> >
> > I only have two (+1) minor remarks.
> >
> > First, in Section 2, it is stated that ```We present the attacks we use in our experiments in Appendix D.1.``` I invite the authors to explicitly mention here that (e.g.) "we define an *attack* as a means to create an adversarial sample".
> >
> > Second, change the very first line of the abstract (which I think can very well be removed -- especially in light of the fact that some works propose adversarial attacks whose samples are indeed "noticed" by a human$^{1,2}$
> >
> > (third: I invite the authors to revise and proof-read the text several times, potentially with the assistance of a native speaker)
> >
> > $^1$ Elsayed. "Adversarial examples that fool both computer vision and time-limited humans." Advances in neural information processing systems 31 (2018).
> >
> > $^2$ Schneider "Concept-based Adversarial Attacks: Tricking Humans and Classifiers Alike." IEEE Security and Privacy Workshops (2022)

---

> > > ### Author Response · Authors · 2022-11-12
> > > **Answer 2 to Reviewer Xhhn**
> > >
> > > Thank you for this further feedback.
> > >
> > > The first paragraph of the related work (section 2) now contains: "Adversarial attacks are algorithms that find such adversarial samples, and they have been particularly successful against neural networks. We present the adversarial attacks we use in our experiments in Appendix D1".
> > >
> > > We have removed the first sentence of the abstract. It was indeed quite unnecessary.

---

> > > > ### Comment · Reviewer_Xhhn · 2022-11-12
> > > > **Thanks**
> > > >
> > > > I thank the authors for acknowledging my suggestion. I have no further remarks on the paper: I did not increase my score (it was already quite high) but I increased my "confidence".

---

> > > > ### Comment · Reviewer_Xhhn · 2022-12-08
> > > > **Question**
> > > >
> > > > Just a question: what is the false positive rate of your approach (i.e., how many "non-adversarial examples are classified as being adversarial examples")?
> > > >
> > > > In Section 5.1, the paper states ```Using its empirical quantiles on clean points allows then to detect samples from some attacks with high recall and a fixed false positive rate```. However, I could not find any mentioning of such fixed false positive rate. Was it measured in any way? And, if yes, can it be gleaned from the provided results?

---

> > > > > ### Author Response · Authors · 2022-12-08
> > > > > **False positive rate**
> > > > >
> > > > > Dear Reviewer Xhhn,
> > > > >
> > > > > Thank you for responding further.
> > > > >
> > > > > 1. "what is the false positive rate of your approach"
> > > > >
> > > > > Since the accuracy on a balanced dataset of clean and adversarial examples is very often superior to 95% (especially in the first table), the false positive rate cannot mathematically be higher than 10% in this case, which is why we did not report it as it is often very low.
> > > > >
> > > > > However, when our code runs, it outputs multiple evaluation metrics (test accuracy, detection rate of successful adversarial samples, false positive rate and true positive rate). So we can provide these results on all the experiments in the paper, and we have already added in the appendix the detection rate of successful adversarial samples for AutoAttack. Here are for example below the false positive rates on ResNeXt50-CIFAR100 (corresponding to the second row in Table 1, averaged over 5 runs). As in the paper, in each cell the first number corresponds to the vanilla network and the second number to the LAP-network. The FPR is always lower on our detector, never higher than 5.5%, and often reduced by the regularization.
> > > > >
> > > > > | Detector - Attack | FGM           | APGD          | BIM           | AA            | DF            | CW          |
> > > > > |-------------------|---------------|---------------|---------------|---------------|---------------|-------------|
> > > > > | TR                | 3.36/**1.88** | 4.64/**2.34** | 2.52/**1.88** | **3.98**/4.06 | **0.24**/0.30 | 6.9/**5.5** |
> > > > > | MH                | 5.22/3.3      | 6.26/3.60     | 4.60/3.30     | 13.74/10.76   | 2.94/2.68     | 7.0/7.2     |
> > > > >
> > > > > Again we can report the FPR for all other experiments.
> > > > >
> > > > > 2. "In Section 5.1, the paper states..."
> > > > >
> > > > > Figure 2 in Section 5.1 is meant as an illustration of how the regularization makes the behavior of the network more different between clean and adversarial examples by looking only at the transport cost (so a single number). This is not meant as a detection method for strong attacks to be used in practice. However, using this quantity (the transport cost) on networks trained with the regularization allows us to detect some weak attacks such as FGM without training any detector or even seeing adversarial samples.
> > > > >
> > > > > The fixed positive rate follows simply from the empirical quantiles we choose for the transport cost on the clean training data. In Figure 2, the vertical lines correspond to the 0.02 and 0.98 quantiles. Considering samples whose transport cost is outside of these quantiles as adversarial will then lead to a false positive rate of around 4% (if there is no distribution shift of course), and the histograms in Figure 2 on the right show that the transport costs of some attacks lie completely outside these quantiles, so the true positive rate is 100%. Again, this is only used for illustration in Figure 2. There is no fixed FPR in the subsequent quantitative experiments (Tables 1 and 2) evaluating the real detection method.

---

> > > > > > ### Comment · Area_Chair_tx2u · 2022-12-09
> > > > > > **Inconsistency?**
> > > > > >
> > > > > > - I guess that  the posted results (second row of Table 1) correspond to  ResNext50/CIFAR100. There the authors report in Table 1 a "detection accuracy" of 84.9% against AA for the non-regularized version of their method. In the rebuttal now they say that they have a FPR of 3.98% at a TPR of 95%. As the problem is described to be balanced, that results in an accuracy of 95.51%. That is inconsistent with the reported accuracy of 84.9% for TR and AA. What is the explanation for this inconsistency?
> > > > > >
> > > > > > - the setting of detecting known attacks in Table 1 is practically of little use. The more relevant setting is detecting an unknown attack, so the FPR would have been interesting for the second row of Table 2. The really interesting setting is to report the FPR when their detector is attacked.
> > > > > >
> > > > > > - the authors write "In the results below, we report the results of the best performing version,
> > > > > > which is often this ensemble of the class-conditional detector and the general detector." What does that mean? That suggests that the authors select for every dataset/attack their best detector from a set of detectors (optimization on the test set). That makes no sense - the detector setting has to be fixed and then only the numbers of the clearly described method have to be reported

---

### Official Review · Reviewer_jGWF · 2022-10-24

**Confidence:** 3
**Correctness:** 3
**Technical Novelty And Significance:** 3
**Empirical Novelty And Significance:** 3
**Recommendation:** 5

**Clarity, Quality, Novelty And Reproducibility:**

The paper is well-written and easy to follow in general. I am not concerned about its originality and reproducibility, after checking the code that's released anonymously by the authors.

**Strength And Weaknesses:**

Strength:
The proposed method has a solid theoretical background and the idea of applying trajectory regularization in optimal transportation to adversarial example detection is also very interesting. The theoretical studies look solid to me. The authors also conducted experiments to show the effectiveness of the proposed method.
Weakness:
1. My first concern is that the authors did not test on adaptive adversaries, which may have access to the detector itself. Similar issues also exist in other works on adversarial examples detection and defense. There are existing works that show that random forest and other decision tree-based models are also vulnerable to adversarial attacks (e.g. Kantechelian et al "Evasion and Hardening of Tree Ensemble Classifiers", ICML16). So I think if the attackers are able to design attacks on the detector, the proposed method can be bypassed.
2. Even for non-adaptive attacks, SOTA attacks such as AutoAttack (Croce and Hein, ICML20) are not tested, either. I think it is necessary to test under strong attacks. For example, LID mentioned in this paper fails under high-confidence adversarial examples according to Athalye et al ICML18.
3. Minor: Please refrain from only using color to distinguish points and bars as in Figure 1, Figure 2 and Figure 3, as it is not friendly to readers with color blindness.

**Summary Of The Paper:**

This paper tries to detect adversarial examples from a perspective that models residual networks as discrete dynamical systems. The detector studies trajectories of samples in space, through time, to distinguish between clean and adversarial examples. Based on this rationale, the authors also apply transport regularization during training to make the detector perform better. Though this method does not necessarily make the model more robust to adversarial attacks, it makes the model's behavior on adversarial examples more distinguishable.

**Summary Of The Review:**

In general, I think this paper proposed an adversarial detection method from an interesting perspective. The theoretical analysis is solid. However, the experimental part may not be enough due to the lack of adaptive attacks and strong attacks, which makes the proposed method less convincing. So I would recommend a borderline reject.

---

> ### Author Response · Authors · 2022-11-18
> **Answer 2 to Reviewer jGWF: AutoAttack**
>
> We address the last two points in the review below.
>
> 2. "SOTA attacks such as AutoAttack are not tested"
>
> Thank you for the suggestion. We have added results using AutoAttack (called AA in the tables) to the paper in all experiments. On all networks/datasets, both when AutoAttack is seen and unseen during detection training, it is much better detected by our detector than by the Mahalanobis detector (by up to 17 percentage points, see Table 1 in Section 5.2 and Table 2 in Section 5.3). It does indeed seem to be more difficult to detect than the other attacks on 2 of the 3 datasets, but its detection improves a lot on networks trained using our transport regularization (by up to 10 percentage points, see Table 1 in Section 5.2, and Tables 7 and 8 in Appendix D6). So the experiments clearly show that our two contributions are very useful for detecting this attack. Also, when looking at detection rates of successful attacks only, detection of AA improves further (almost 100% in Table 10 in Appendix D7 when AutoAttack is seen during detection training).
>
> Please note that the version of PGD we use is already Auto-PGD-CE as seen in the released code, which is already one of the four attacks that form AutoAttack. We added this to the paper with the proper name and citation.
>
> Also, (Athalye et al, ICML18) say that they generate the "high confidence adversarial samples" that fool the LID detector through the $L_\infty$ CW attack. We already use the $L_2$ version of CW, which is to the best of our knowledge considered one of the strongest attacks available.
>
> 3. "Please refrain from only using color to distinguish points and bars as in Figure 1, Figure 2 and Figure 3"
>
> We apologize for the inconvenience and have redone all 4 figures in the paper using shapes and color-blind-friendly color schemes to distinguish points and bars. Figure 1 now uses both shapes and a color-blind-friendly color scheme, but unfortunately, because of the scale of the experiment, the points are too small for their shapes to be visible. The other figures should all be readable in black-and-white.

---

> ### Author Response · Authors · 2022-11-18
> **Answer 1 to Reviewer jGWF: Adaptive Attacks**
>
> Thank you very much for your review. Please note that there have been changes to the paper:
> - Tables 1 and 2 and Sections 5.2.1 and 5.2.2 (white-box and black-box attacks) have been merged into Section 5.2 and Table 1.
> - A comparison with RCE (a training method that aims to improve detectability of adversarial samples) has been added to the appendix. We verify that this method does often improve detection, but less than our regularization.
> - Experiments on AutoAttack, adaptive attacks and OOD detection have been added. The code in the github has been updated.
>
> We address the points you raise below.
>
> 1. "My first concern is that the authors did not test on adaptive adversaries, which may have access to the detector itself."
>
> We have tested 2 attacks on our detector and the Mahalanobis detector in Section 5.5, with details in Appendix D10. Both are white-box with respect to the network. The first is black-box with respect to the detector and only knows if a sample has been flagged as adversarial or not. The second has some knowledge about the detector. It knows what features it uses and can attack it directly to find adversarial features. We test these attacks by looking at the percentage of detected successful adversarial samples (i.e. that fool the network) that they turn into undetected successful adversarial samples.
>
> - The 1st attack proceeds as follows. A strong white-box attack (CW or AutoAttack) is used on the network on image $x$ that has label $y$. If it finds a successful adversarial image $\tilde{x}$ that fools the network into predicting $\tilde{y} \neq y$ and is detected by the detector, the attacker will attempt to modify this image $\tilde{x}$ so that the network and the detector are both fooled. For this, the image $\tilde{x}$ is used as the initialization for an attack (HSJ) on a black-box Network-Detector. The attacker considers that the Network-Detector behaves as follows: it outputs the class prediction of the network if the detector does not detect an attack and outputs an additional `detected' class if the detector detects an attack. The attacker attacks the Network-Detector on $\tilde{x}$ targeting the $\tilde{y}$ label. This way the network makes a mistake and the 'detected' class is avoided. On a LAP-ResNet110, this attack turns 6.8% of 1700 detected successful adversarial samples into undetected successful adversarial samples on our detector, and 12.9% on the MH detector, and these percentages are lowered by LAP training compared to vanilla training. So our LAP training improves the robustness of both detectors, and our detector is more robust than the MH detector.
>
> - The 2nd attack is similar to the adaptive attack used in Carlini & Wagner (2017b) to break the KD detector of Feinman et al. (2017). A strong white-box attack is used on the network on image $x$ that has label $y$. If it finds a successful adversarial image $\tilde{x}$ that fools the network but is detected by the detector, the detection features $\tilde{z}$ of $\tilde{x}$ are used as the initialization for an attack (HSJ) on the detector. If successful adversarial detection features $z^*$ that fool the detector are found, the attacker has to find an image that still fools the network and that has these detection features $z^*$ (or close features that fool the detector). We do this similarly to Carlini & Wagner (2017b) by solving:
> $$\min_{x^*} - L(N(x^*), y) + c_1 \| D(x^*) - z^* \| + c_2 \| x^* - x \|$$
> where $L$ is cross-entropy, $N$ is the network, and $D$ returns the detection features of its input. This optimization problem is differentiable and we try differentiable algorithms such as BFGS to solve it. As in Carlini & Wagner (2017b), $\tilde{x}$ is used as initialization. This attack turns 14% of detected successful adversarial samples into undetected successful adversarial samples on our detector.
>
> Given that initial detection rates of successful adversarial samples are almost 100% (Table 10 in Appendix D7), this shows that adaptive attacks do not (at least not easily) circumvent the detector, as detection rates drop to 85% at worst. The second attack is stronger than the first one, but it can probably still be improved by using a white-box attack that is specific to random forests such as the one you cite for attacking the detector, or another loss than cross entropy such as the CW one. But the difficulty of combining the attack on the network with that on the detector remains. It is the non-differentiability of the random forest that forces either this separate treatment of network and detector or the use of a black-box method on both as in the 1st attack. Also, we did not consider here the ensembled class-conditional version of the detector (Section 5.2), which is the best performing one and should be more robust to adaptive attacks, as the attacker has to fool 2 random forests at once and target a particular label, constraining further the optimization problem he solves. Please see App D10 for more details.

---

### Official Review · Reviewer_J6gT · 2022-10-24

**Confidence:** 2
**Correctness:** 3
**Technical Novelty And Significance:** 3
**Empirical Novelty And Significance:** 2
**Recommendation:** 5

**Clarity, Quality, Novelty And Reproducibility:**

I found the clarity of the paper overall lacking. The paper introduces what I believe to be a novel detection framework, and a novel application of the particular regularization method.

**Strength And Weaknesses:**

Strengths
- The authors show their detection method performs better than the Mahalanobis detector on known and unknown white box attacks, and also on known black-box attacks.
- The regularization method proposed also improves the performance of the existing Mahalanobis detector.
- The authors give theoretical backing behind their choice of regularization method.

Weaknesses
- Overall I found the paper hard to follow. I think the paper would benefit from more explicitly stating how the detection is implemented.
- The method is restricted to residual networks, whereas the detector being compared to (the Mahalanobis detector) I believe is applicable to different types of architectures.
- I'm not sure whether you specify that you are considering the l-infinity norm threat model in section 5.
- There are a lot of unclear statements and loose ends. For example, rather than saying "We don’t fine-tune these hyper-parameters much", it'd be better to be more specific with respect to the details.
- There was never much of an introduction or review of adversarial examples more formally -- you may wish to add this.
- The computational complexity of the detection method was not made explicit.
- You could more explicitly compare your method to the method of comparison (the Mahalanobis detector) in the related works section for better clarity.

Other comments/questions:
- Why do you use the test set for training the detector? It seems like it would be more reasonable to hold out part of the training set.
- This paper focuses on detecting adversarial examples specifically, how does it compare to existing methods on other OOD detection tasks?

**Summary Of The Paper:**

This paper proposes an adversarial example detection mechanism for residual networks, as well as a regularization method to improve the detection performance. The detection mechanism is based upon viewing the deep network as a dynamical system. The regularization method is an existing method based on optimal transport that the authors show theoretically and empirically makes adversarial examples easier to detect.

**Summary Of The Review:**

I am leaning towards reject because I found the paper lacking clarity/difficult to follow in terms of the methods proposed. However, the final results of the paper are promising.

---

> ### Author Response · Authors · 2022-11-10
> **Answer 1 to Reviewer J6gT: Further explanation of the method**
>
> Thank you very much for your review. We will try to make the idea more clear below. We have also modified the paper (changes are in blue) to include these clarifications and OOD experiments. The code in the anonymous github has been updated to include the new experiments.
>
> Please note that tables 1 and 2 and sections 5.2.1 and 5.2.2 (white-box and black-box attacks) have been merged into table 1 and section 5.2 to provide more space for this. Also a comparison with RCE (a training method that aims to improve detectability of adversarial samples) has been added to the tables in the appendix. We verify that this method does often improve detection, but clearly less than our method. We also add the Auto Attack requested by another reviewer.
>
> First we would like to point out, in relation to the first point in your review (under strengths), that we do test our method on unknown black-box attacks in section 5.3.
>
> We now address the points you raise.
>
>
> 1. "Overall I found the paper hard to follow. I think the paper would benefit from more explicitly stating how the detection is implemented":
>
> Adversarial detectors often work as follows: an image is fed through the trained network and the activations/embeddings of this image are extracted. Since they have a very large dimension, they are processed to extract features that the method deems useful for detection (this is in the second paragraph in the introduction). The features from clean and adversarial images are used to train a binary classifier that tells clean from adversarial images.
>
> We simply train a random forest with the norms and cosine similarities (to a fixed vector of ones) of the residuals as features and as labels whether the image is adversarial or not. So given an input $x_0$ that goes through a network that applies $x_{m+1} = x_m + h \ r_m(x_m)$ for $0 \leq m < M$ (before classifying $x_M$ by a linear layer), the feature vector given to the random forest is
> $$
> (  \frac{1}{d_1}\|r_1(x_1)\|^2,...,\frac{1}{d_{M-1}}\|r_{M-1}(x_{M-1})\|^2, \ \cos (  r_1(x_1), 1_1 ),..., \ \cos ( r_{M-1}(x_{M-1}), 1_{M-1})  )
> $$
> and the label is 0 if $x_0$ is clean and 1 if it is adversarial. Here $\cos$ is the cosine similarity between two vectors and $1_m$ is a vector of ones of size $d_m$ where $d_m$ is the size of $r_m(x_m)$ which is flattened and treated as a vector. We added this with more detail to the paper in Section 4.1 and equation (7).
>
>
> 2. "The method is restricted to residual networks, whereas the detector being compared to (the Mahalanobis detector) I believe is applicable to different types of architectures":
>
> The method is not theoretically nor implementation-wise restricted to residual networks. As we mention at the end of section 4.2.1, the analysis and implementation still apply to any network on layers that keep the same input and output dimension and so for which a residue can be computed and which can therefore be re-written as residual blocks (through $r_m(x) = g_m(x_m) - x_m$ if the layer is $x_{m+1} = g_m(x_m)$). The same analysis will then immediately follow on these layers. The presentation is done for resnets for this reason, as other networks will have to be reformulated as resnets anyway. We included this brief discussion in section 4.1.
>
> In practice, all the networks we use (other than in the toy example in Figure 1) are already outside the strict theoretical framework, as they include some downsampling layers that change the dimension and are therefore ignored by the detector and the regularization, and as the ResNeXt model is not an Euler scheme as the activation is applied after the shortcut (i.e. $x_{m+1} = \text{ReLU}(x_m + h \ r_m(x_m))$). And yet the method works very well on all three networks. Also note that many different networks such as EfficientNet and MobileNet are made up in large part of of residual blocks, and the method can be immediately applied to them. This is mentioned in Section 4.1 and in the introduction.
>
>
> 3. "I'm not sure whether you specify that you are considering the l-infinity norm threat model in section 5.":
>
> We considered the 2-norm for CW and HSJ and the infinity-norm (the most commonly used) for the other attacks. We added this to the first paragraph of section 5. Other than the parameters mentioned, we use the default values in the ART package we use to generate the adversarial samples.
>
> 4. "There are a lot of unclear statements and loose ends. For example, rather than saying "We don’t fine-tune these hyper-parameters much", it'd be better to be more specific with respect to the details."
>
> We give the values used for all 3 hyperparameter of the regularization in the second paragraph in Section 5. All three are equal to 1 for all networks and datasets and they lead both to good classification accuracy and good adversarial detection. So indeed the hyperparameters do not require much finetuning. And they are only chosen during training of the network, so not to improve adversarial detection.

---

> ### Author Response · Authors · 2022-11-10
> **Answer 2 to Reviewer  J6gT: Further details and OOD experiments**
>
> We continue below answering the points raised in the review. Please see the previous answer for the first 4 points, and the updated version of the paper (changes are in blue).
>
> 5. "There was never much of an introduction or review of adversarial examples more formally -- you may wish to add this.":
>
> We present the attacks we use in appendix D1 because of the length limit. We added a definition of adversarial samples in the first paragraph of the related work (section 2): given a classifier $f$ in a multi-class classification task and $\epsilon > 0$, an adversarial sample $y$ constructed from a clean sample $x$ is $y = x + \delta$, such that $\| \delta \| \leq \epsilon$ and $f(y) \neq f(x)$. The maximal perturbation size $\epsilon$ has to be so small as to be almost imperceptible to a human.
>
>
> 6. "The computational complexity of the detection method was not made explicit."
>
> There is a favorable time comparison of our detector with the Mahalanobis (Lee et al. 2018) detector in appendix D8. We added to it the computational complexity which is simply that of computing a norm and a scalar product per block so O(MD) where M is the number of blocks and D the largest embedding dimension. Training a random forest on these features takes only a few seconds as mentioned in appendix D8.
>
> 7. "You could more explicitly compare your method to the method of comparison (the Mahalanobis detector) in the related works section for better clarity."
>
> We mention in the second paragraph in the related work how the Mahalanobis (Lee et al. 2018) detector operates (it uses Mahalanobis distances between the activations and a Gaussian fitted to them during training, assuming therefore that they are normally distributed). We have now added after this description a sentence describing the main difference between our method and the Mahalanobis detector. It is that our method does not come from a statistical approach and does not require the extraction of statistics from the network's training set nor batch-level statistics, whereas Mahalanobis needs statistics from the network's training set. The comparison between different samples happens only during the training of the random forest on the features. The features themselves are computed from each individual image alone. All in all, our detector is quite original as it is not inspired by statistical or computer vision considerations. The fact that it is lightweight (only two features per block that are quick to calculate) allows us to consider all blocks, whereas many methods have to keep features from only a few blocks to save memory and/or time.
>
>
> 8. "Why do you use the test set for training the detector? It seems like it would be more reasonable to hold out part of the training set"
>
> The network is trained on the training set, the test set is separated into two subsets. One is used to train the detector and the second for evaluating the detector's performance. This provides an unbiased estimate of the error.
>
> In the paper, train set and test set refer to the ones provided by PyTorch. Training on the entire train set allows to have networks trained to close to state-of-the-art accuracy level and to test whether the regularization improves both classification test accuracy and adversarial detection accuracy at the same time in a normal training regime on the full train set. Note that Mahalanobis (Lee et al. 2018) state that they use part of the test set for training the detector, and cite two other detectors including LID (Characterizing Adversarial Subspaces Using Local Intrinsic Dimensionality, ICLR 2018) that do the same. So this does seem standard.
>
>  When a validation set is also provided by PyTorch (TinyImageNet), it is indeed the one we use for training the detector.
>
>
> 9. "This paper focuses on detecting adversarial examples specifically, how does it compare to existing methods on other OOD detection tasks?"
>
> Thank you for the suggestion. Indeed our analysis applies to all OOD samples and so we might want to test the method on detection of OOD samples. However there are many tasks and settings in OOD detection.
>
> We add a section 5.4 at the end of the experiments section (with details in Appendix D9) containing an experiment in a similar setting to the one used in the Mahalanobis paper (Lee et al. 2018) for OOD detection: we train a network on a first dataset (CIFAR10), then we train detectors to distinguish between this first dataset and a second dataset (CIFAR100 or adversarially perturbed CIFAR10), and then test their ability to distinguish between the first dataset and a third unseen dataset (SVHN). Our detector does very well and better than the Mahalanobis detector in both experiments we run. Detection of unseen OOD samples reaches more than 90% accuracy with our detector. So without any extra data available, using an adversarial attack allows to detect OOD samples from an unseen distribution with more than 90% accuracy.

---

### Decision · Program_Chairs · 2023-01-20

**Decision:**

Reject

**Justification For Why Not Higher Score:**

The experiments are lacking in many aspects as explained above and the updated experiments for unseen AutoAttack suggest that the detector is worse than the initially reported experiments suggest. The field of detection of adversarial examples has seen many cases where published claims were shown later to be wrong (see papers by Carlini et al). Thus it is even more important to maintain high standards in the experiments.

**Justification For Why Not Lower Score:**

N/A

**Metareview: Summary, Strengths And Weaknesses:**

Two reviewers were leaning towards reject, one reviewer has been very positive about the paper. The pros and cons were extensively discussed.

The points raised by the reviewers were:
- unclear threat model
- no adaptive attacks
- stronger attacks (AutoAttack) not tested
- connection between theory and actual algorithm unclear

The reviewers liked the theoretical contribution regarding trajectory regularization to detect adversarial examples.

The authors have invested quite some time in the rebuttal to fix these issues which I appreciate. However, I think that there are still some open points left:

- no threat model: even in the updated version at the beginning of Section 2 no clear threat model is stated, the changes are norm-bounded, but the norm is not specified. It remains unclear if the authors claim robustness against any norm-bounded attack or l_p-attacks in general or just the $l_\infty$-attacks and $l_2$-attacks done in the experiments (even though the radius for the $l_2$-attacks is nowhere specified). Even if one just considers $l_\infty$-attacks: do the authors claim robustness against a specific radius or any $l_\infty$-attack? (see my comments below on unseen attacks)

-  no adaptive attacks: the paper has added some adaptive attacks but none of them is end-to-end (which is also difficult as there is no threat model). The authors use in the updated version a black-box/grey box setting but a complete white-box attack setting is needed (which is likely to break the detector as shown in many papers by Carlini et al) to check the limitation of the system followed by an end-to-end black box attack. Also the updated version does not state which attack is used (CW or AA?) and the budget of the attacker (e.g. iterations for PGD and queries for the black box attack) which makes it irreproducible and it remains unclear how much effort has been invested in breaking the detector.

- unseen attacks: the authors should use attacks which try to search for minimal norm perturbations on the decision boundary as they are the most difficult attack for their detector setting. The current attacks are mainly finding points on the boundary of the threat model as they maximize the loss. In the updated version the AutoAttack examples seem more difficult to detect (which in my point of view shows that the detector does not work properly). As AutoAttack consists of four attacks and the Fast Adaptive Boundary (FAB) attack is one of them, which tries to find points on the decision boundary, this hints on that such attacks are much more difficult to detect for the detector.


- no proper detection setting: the paper reports always "detection accuracy" but a useful detector in reality needs to pass most unattacked examples and only reject adversarial examples. Thus one uses typically FPR at a fixed TPR (often 95%, but other papers have even used 99%) as a measure (here positives corresponds to unperturbed examples). Reporting accuracy is useless to judge the utility of the detector. It was not clear what numbers were reported in the last answer of the authors  - it should be clearly defined in the paper. Reference to the code is not a replacement for a correct description in the text.

In total the paper in its current form is not ready for publication and requires a major revision.

Minor points:

 -  the accuracy of the trained networks is very low e.g. only 92.5% for CIFAR10, but there exists on RobustBench already an adversarially robust model with an accuracy of 92.2% and having a robust accuracy of 66.5% against $l_\infty$ of radius 8/255, or a model with 95.7% accuracy and having 82.3% robust accuracy against $l_2$-attacks of radius 0.5. These models could serve as a baseline even so they don't use a detector

- the model selection of the detector has to be clearly explained or better only one a fixed detector setting should be reported